# Oligoprotein type I interferon signatures, but not *TREX1* variants, increase risk of systemic lupus erythematosus in UK Biobank

Bastien Rioux [1,2], Sarah McGlasson[1,2], Deborah Forbes[1,2], Katy R. Reid[1,2], Anna Klingseisen[1,2], Joe Berry[3], Neeraj Dhaun[4], Wan Fai Ng[3,5,6], William Whiteley [1,7] & David P. J. Hunt [1,2] ✉

The 3′−5′ DNA exonuclease, TREX1, is a negative regulator of the type I interferon response, while *TREX1* variants are considered to confer risk for non-monogenic systemic lupus erythematosus (SLE). Here we analyse *TREX1* sequences in 469,229 UK Biobank participants together with multi-omics data from the UK Biobank Pharma Proteomics Project to reappraise the contribution of reported *TREX1* risk variants in SLE. We find that *TREX1* variants are not associated with increased risk for SLE in UK Biobank, and most reported risk variants are functionally neutral in mutagenesis experiments. Deriving an oligoprotein interferon signature from broad capture proteomics, we find that this signature is associated with elevated SLE risk, but is not elevated in *TREX1* variant carriers. Furthermore, *TREX1* variants are not associated with other autoimmune diseases with a prominent oligoprotein interferon signature. Finally, meta-analysis of published studies confirms the lack of support for the association between SLE and *TREX1* risk variants. In summary, we find that, while oligoprotein type I interferon signatures increase risk of SLE, *TREX1* variants do not.

TREX1 is the major 3′–5′ human exonuclease, which degrades ssDNA/dsDNA and, through removal of self-nucleic acids, acts as a negative regulator of the type I interferon response to prevent autoinflammation and autoimmunity[1]. TREX1 is a dimer, and reduced exonuclease activity in humans is most commonly due to biallelic loss of function variants[2], leading to an autosomal recessive condition called Aicardi–Goutières syndrome (AGS), characterised by widespread activation of the type I interferon response[3] and manifestations that can include features of systemic lupus erythematosus (SLE). Parents of children with AGS appear to produce increased levels of antinuclear antibodies[4], and dominant variants in *TREX1* can cause a monogenic cutaneous form of lupus known as familial chilblain lupus (FCL)[5].

Such a link between *TREX1* variants and lupus-like conditions is mechanistically plausible given that SLE is an autoimmune disease associated with anti-nucleic acid antibodies and type I interferon activation[6]. Candidate gene studies have specifically tested this hypothesis of a link between *TREX1* variants and SLE. The first study[7] to do so reported an association of ten coding *TREX1* variants with SLE or Sjogren disease (SjD). These variants conferred a 25-fold increased risk of developing SLE. This association between *TREX1* and manifestations of SLE was considered to be confirmed following a large candidate gene study[8], contributing to the general acceptance that *TREX1* variants predispose to non-monogenic SLE, which includes annotation of variants in clinical reference databases[9–13]. As such, the association

[1]Institute for Neuroscience and Cardiovascular Research, University of Edinburgh, Edinburgh, UK. [2]UK Dementia Research Institute at Edinburgh University, Edinburgh, UK. [3]Translational and Clinical Research Institute, Newcastle University, Newcastle upon Tyne, UK. [4]Edinburgh Kidney Research Group, University/BHF Centre for Cardiovascular Science, The Queen's Medical Research Institute, University of Edinburgh, Edinburgh, UK. [5]University College Cork, Cork, Ireland. [6]Department of Rheumatology, Cambridge University Hospitals, Cambridge, UK. [7]British Heart Foundation Data Science Centre, Health Data Research UK, London, UK. ✉e-mail: david.hunt@ed.ac.uk

between *TREX1* variants and strong risk of SLE is widely considered as valid. However, with the development of more powerful tools to dissect genetic associations with immune diseases and associated endotypes, including the potential to measure markers of type I interferon activation at scale, it is important to re-evaluate what are considered to be canonical associations.

UK Biobank (UKB) is a large population-based study of 502 k volunteers recruited through UK patient registries[14]. It offers a number of features which allow this genetic association to be tested in detail, including *TREX1* exome sequencing in ~469 k participants and reliable capture of SLE diagnoses through routinely collected healthcare data[15,16]. Moreover, UKB participants were aged 40–69 years at enrolment in 2006–2010, and the study captures both incident and prevalent diagnoses, meaning that most participants predisposed to SLE will by now have been diagnosed, as SLE incidence peaks in middle age[17]. Since elevated type I interferon is an important feature of SLE, blood proteomics generated in the UKB Pharma Proteomics Project (UKB-PPP) in ~54 k participants also offers an opportunity to develop and detect oligoprotein interferon signatures, which form an important part of the immune endotypes associated with SLE. UK Biobank is therefore a powerful multi-omics resource to evaluate the *TREX1*-SLE association.

In this study, we re-examine the association between rare coding *TREX1* variants and SLE. We begin by showing that previously reported *TREX1* risk variants are largely neutral in functional studies and are not associated with SLE or SjD risk in UKB. We next develop and validate a type I interferon signature score using blood proteomics in UKB-PPP to define a cluster of type I interferon-related autoimmune conditions, and show that *TREX1* variants are not associated with such type I interferon signature or interferonopathic autoimmunity. We finally conduct a systematic review and meta-analysis of published *TREX1*-lupus association studies, which does not support a contribution of *TREX1* variants in SLE. Together, these findings suggest that rare coding *TREX1* variants are not associated with type I interferon dysregulation and do not confer risk for SLE or other type I interferon-related autoimmunity.

## Results

### No association between *TREX1* variants and SLE in UK Biobank

We first aimed to replicate results from a previous genetic association study[7] linking monoallelic *TREX1* variants to SLE. Using a set of 10 *TREX1* risk variants found in SLE or SjD cases but not in controls from this study, we identified 866 carriers among 469,229 UKB participants (0.18%) with whole-exome sequencing after sample and variant quality control (Fig. 1A, B). Three variants had allele counts ≤2 (c.635del [P212Hfs], c.812_813insAA [D272Rfs] and c.917G > C [G306A]), whereas c.341G > A (R114H) accounted for 619 (71.5%) of *TREX1* variant carriers. There was no homozygote for minor alleles and one variant (c.739G > C [A247P]) was always found in compound heterozygotes with c.679G > A (G227S). Carriers, as compared to non-carriers, had similar age and sex distributions but a higher proportion of non-European ancestry (27.0 vs. 16.1%; see Supplementary Table 1 for details).

Before testing the association of these *TREX1* variants with SLE in UKB, we performed recombinant protein studies in *Trex1*$^{-/-}$ mouse embryonic fibroblasts (MEFs) to examine their impact on exonuclease function and localisation. As expected, the two frameshift variants (P212Hfs and D272Rfs) affected protein localisation while the remaining variants had no impact on TREX1 localisation. The nuclease activity assay showed that only the two AGS-causing variants (R114H and P212Hfs) had any effect on nuclease activity, while the remaining variants had no effect on nuclease function (Fig. 1C, D; see Supplementary Fig. 1 for details).

We next identified 1306 UKB participants with SLE, of which most (69.5%) had at least one diagnostic code from hospital inpatient admissions data (Fig. 2A; see Supplementary Table 2 for details).

Participants with SLE were most frequently diagnosed between 40 and 60 years old (42.9%) and, compared to those without SLE, were more frequently females (82.8 vs. 54.1%) and of non-European ancestry (24.9 vs. 16.1%; see Supplementary Table 3 for details). The prevalence of *TREX1* variants among people without SLE was higher in UKB (0.19%) than in previously reported cohorts[7] (Fig. 2B). None of the carriers were diagnosed with SLE in UKB, yielding a neutral association between this set of *TREX1* risk variants and SLE with confidence intervals (CIs) that exclude a clinically meaningful effect (odds ratio [OR] = 0.178; 95% CI: 0.001, 1.207; *p* = 0.090; Fig. 2C). Peripheral markers of type I interferon activation are also observed in SjD[18], and prior candidate gene analyses have also included patients with either SLE or SjD[7]. Combining SLE and SjD to define cases as previously done because of a common link with type I interferon[7] (see Supplementary Table 4 and Supplementary Fig. 2 for sources of diagnosis) yielded similar neutral findings, with a single case among carriers (0.1%; heterozygote for c.341G > A [R114H]) as compared to 2642 cases among non-carriers (0.6%). Comparable results were obtained after excluding self-reports and for SjD only (Fig. 2C). These neutral findings were also reproduced for European ancestry participants (*n* = 393,433) using burden tests and optimal sequence kernel association tests (SKAT-O) implemented in SAIGE-GENE+[19] using weights based on minor allele frequency (MAF) and the inverse of residual enzymatic function from our in vitro exonuclease assays (Supplementary Table 5).

### An oligoprotein interferon signature identifies interferonopathic autoimmune diseases

SLE is considered the prototypical example of a sporadic autoimmune disease associated with prominent activation of the type I interferon response[6]. Early studies linking *TREX1* with SLE also included conditions such as SjD which is similarly associated with elevated type I interferon. This is biologically plausible since TREX1 is a negative regulator of the type I interferon response. We therefore asked whether we could use UKB-PPP to identify interferonopathic autoimmune disorders and test their association with *TREX1* mutations. To date, however, there has been no systematic identification of autoimmune disorders associated with activation of the type I interferon response. This is because quantification of type I interferon has been challenging due to the low abundance of these proteins. Measuring interferon-α concentrations requires high sample volumes, and for this reason is challenging to measure at scale in large cohort studies such as UKB. An alternative surrogate for interferon-α protein concentrations is RNA-based interferon signatures, which are often used as a biomarker in clinical studies of interferonopathic diseases[20]. However, transcriptomic data are not currently available for UKB.

To overcome these challenges, we sought to develop and validate an oligoprotein signature of type I interferon activation derived from broad-capture proteomics measured at recruitment in UKB. We included 22 correlates of type I interferon levels available in UKB-PPP (Olink Explore 3072) in a penalised logistic regression model trained to achieve the best cross-sectional prediction of recombinant type I interferon therapy (interferon-β in people with multiple sclerosis at the time of baseline blood sampling, Fig. 3A). The oligoprotein interferon signature, which we termed the Markers of type I Interferon Response in Olink (MIRO) score, was then derived as the sum of the three proteins retained in the model (SIGLEC1, RIG-I and IFIT3) weighted by their penalised coefficient followed by standardisation. We tested the score in three validation experiments. Firstly, the oligoprotein interferon signatures were strongly increased in individuals treated with recombinant interferon-α at the time of blood sampling (Fig. 3B). Secondly, MIRO scores had the strongest correlation with proteins involved in the type I interferon response across other analytes available in UKB-PPP (Fig. 3C). Thirdly, we externally validated the score in a cohort of individuals with SjD from the United Kingdom Primary Sjögren's Syndrome Registry (UKPSSR), where paired interferon-α single molecule

**A**

| cDNA change | Protein change | Carriers | Minor allele frequency (%) | | | CADD | Effect | ClinVar |
|---|---|---|---|---|---|---|---|---|
| | | | UKB | gnomAD | 1000GP | | | |
| c.341G>A | p.Arg114His | 619 | 0.0660 | 0.0208 | . | 29.3 | Missense | P/LP |
| c.473C>T | p.Ala158Val | 67 | 0.0071 | 0.0044 | . | 23.8 | Missense | VUS |
| c.635del | p.Pro212HisfsTer65 | 1 | 0.0001 | 0.0004 | . | 28.9 | Frameshift | P/LP |
| c.679G>A | p.Gly227Ser | 69 | 0.0074 | 0.0127 | 0.0400 | 7.1 | Missense | VUS |
| c.720G>C | p.Arg240Ser | 51 | 0.0054 | 0.0203 | 0.0800 | 5.1 | Missense | Conflicting (VUS/LB) |
| c.739G>C | p.Ala247Pro | 62 | 0.0066 | 0.0119 | 0.0400 | 22.9 | Missense | VUS |
| c.812_813insAA | p.Asp272ArgfsTer6 | 0 | . | . | . | 23.0 | Frameshift | NA |
| c.869C>T | p.Pro290Leu | 40 | 0.0043 | 0.0064 | . | 21.9 | Missense | VUS |
| c.914A>G | p.Tyr305Cys | 17 | 0.0018 | 0.0120 | . | 25.5 | Missense | VUS |
| c.917G>C | p.Gly306Ala | 2 | 0.0002 | 0.0004 | . | 18.4 | Missense | VUS |

**B**

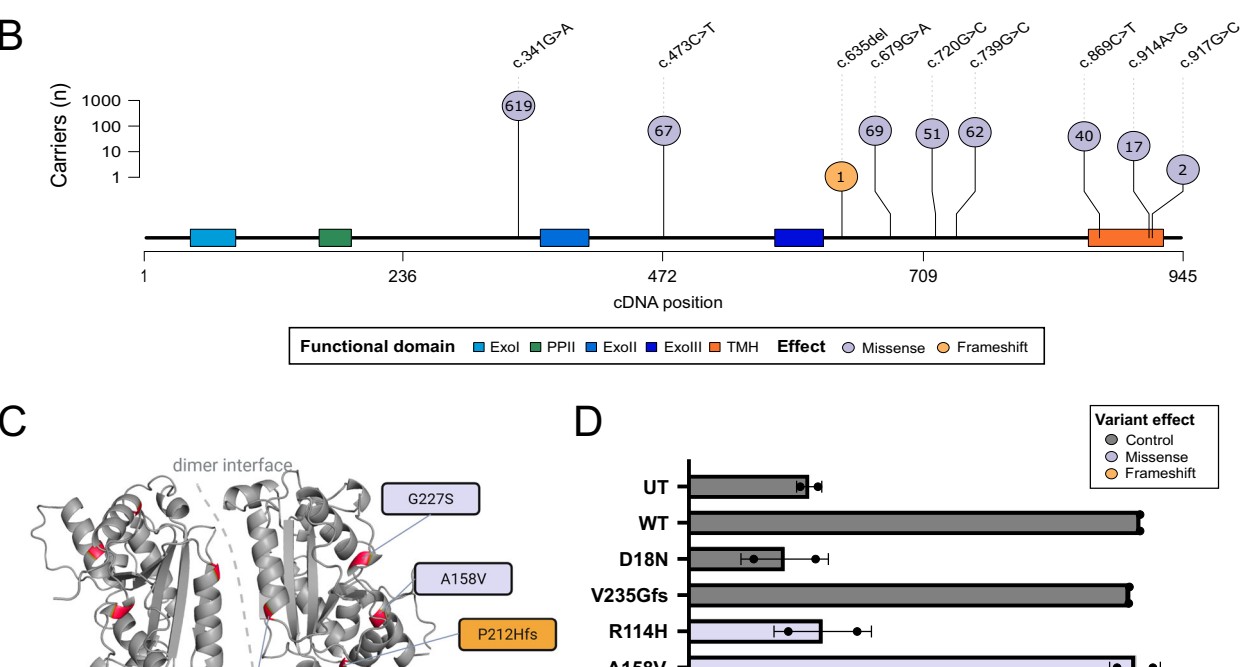

**C** **D**

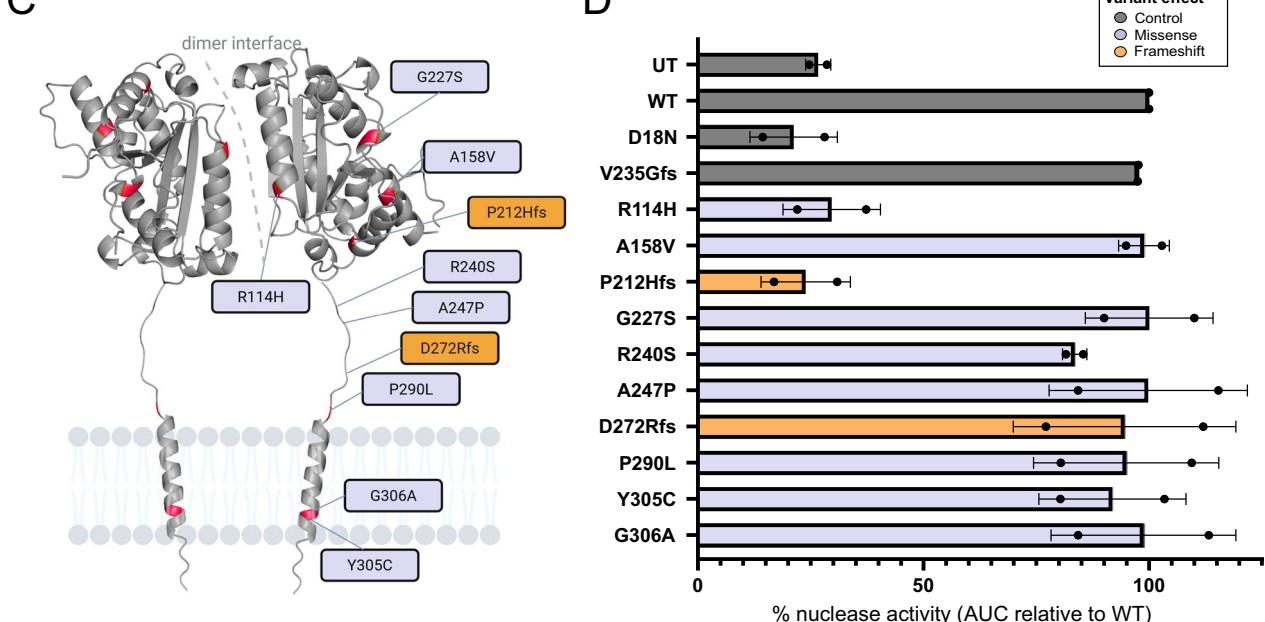

ELISA, Olink proteomics and expression data were measured in blood. We confirmed that components of the MIRO score are interferon-stimulated genes (ISGs) and found a significant correlation between the oligoprotein interferon signature and interferon-α concentrations, but not interferon-γ (Supplementary Fig. 3).

We next used this oligoprotein signature to survey type I interferon activation across 21 common autoimmune diseases (see

Supplementary Tables 6 and 7 for definitions). Ten autoimmune conditions were associated with a high ($\beta > 0.6$: SjD, systemic sclerosis, dermatomyositis, SLE) or moderate ($\beta \geq 0.2$ and $\leq 0.6$: primary biliary cirrhosis, coeliac disease, vasculitis, vitiligo, rheumatoid arthritis and type 1 diabetes) type I interferon signature (Fig. 3B). Eleven other conditions including multiple sclerosis were only weakly or not associated ($\beta < 0.2$) with such signature.

**Fig. 1 | Characterisation of reported *TREX1* SLE risk variants in UK Biobank.**
**A** Description of reported *TREX1* SLE risk variants. ClinVar refers to variant classifications from individual submitters for type I interferonopathies (excluding retinal vasculopathy with cerebral leukoencephalopathy [RVCL]) as: P, pathogenic; LP, likely pathogenic; VUS, variant of uncertain significance; LB, likely benign.
**B** Lollipop plot of carrier counts for reported *TREX1* SLE risk variants in UK Biobank. Source data are provided as a Source Data file. **C** 3D rendering of TREX1 with SLE risk variants mapped. **D** Impact of *TREX1* SLE risk variants on enzymatic function in *Trex1*[-/-] MEFs. The AUC was measured and normalised within each experiment to

WT = 100%. Bars represent SD from $n = 2$ independent experimental replicates (separate transfections). Dots show the values of area under the curve for each experimental replicate. Source data are provided as a Source Data file. 1000GP 1000 Genomes Project, AUC area under the curve, CADD Combined Annotation-Dependent Depletion score (PHRED-like scaled), cDNA coding DNA, Exo exonuclease region, gnomAD Genome Aggregation Database exome, MEFs mouse embryonic fibroblasts, PPII polyproline II motif, SLE systemic lupus erythematosus, TMH transmembrane helix, UKB UK Biobank, UT untreated, WT wild type.

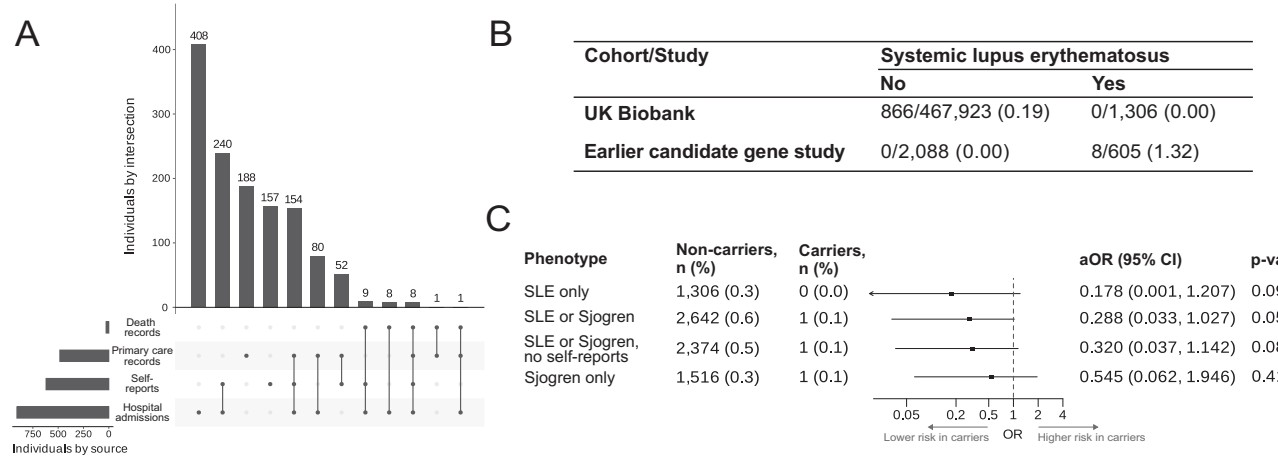

**Fig. 2 | Previously reported *TREX1* SLE risk variants are not associated with SLE in UK Biobank. A** UpSet plot of sources of diagnosis for SLE in UK Biobank. **B** Carriers of *TREX1* SLE risk variants by disease status in UK Biobank and an earlier genetic association study[7]. Numbers refer to carriers / total (%). **C** Clinical associations of previously reported *TREX1* SLE risk variants[7] in UK Biobank. Associations

examined using logistic regressions with Firth penalisation adjusted for age, sex and the first ten genetic principal components (two-sided tests). Data are presented as adjusted OR (95% CI). aOR adjusted odds ratio, CI confidence interval, SLE systemic lupus erythematosus.

## *TREX1* variants are not associated with interferonopathic autoimmunity

We used this oligoprotein interferon signature approach to test a broader hypothesis that carriers of any disease-causing *TREX1* variants have an increased risk of type I interferon-related disease. We first expanded the set of *TREX1* variants reported to cause AGS, FCL or autoimmunity from a systematic search of ClinVar[21], the Leiden Open source Variation Database (LOVD)[22] and two reviews[9,20], which yielded 84 variants involved in monogenic conditions (AGS and FCL) or autoimmunity (Fig. 4A; see Supplementary Table 8 for full description). Variants linked to autoimmunity all cited a previous genetic association study[7]. We found 35 of these 84 variants in UKB (41.7%; Fig. 4B), yielding 1,216 carriers (all heterozygotes with one compound heterozygote [c.553C > T + c.739G > C]). These carriers of reported disease-causing *TREX1* variants did not have an increased risk of any type I interferon-related autoimmune disease (Fig. 4C). Given prior reported associations between neurological manifestations of lupus and *TREX1* variants[8], we also examined associations between *TREX1* variants and six neurological comorbidities of SLE. No association was found (Fig. 4C; see Supplementary Fig. 4 for all individual traits tested), and imaging-derived brain volumes associated with SLE[23,24] did not differ by carrier status (Fig. 4D).

## Oligoprotein interferon signatures are associated with increased risk of SLE, but are not elevated in *TREX1* mutation carriers

Serological evidence of SLE can precede diagnosis by many years[25]. Indeed, in UKB-PPP, we found that elevated type I interferon signatures at baseline in people without an SLE diagnosis were strongly associated with future risk of SLE diagnosis (Fig. 5A; see Supplementary Table 9 for all estimates). Oligoprotein interferon signatures could be detected up to 9 years prior to diagnosis (Supplementary Fig. 5). We therefore

explored whether *TREX1* variant carriers might have evidence of peripheral type I interferon activation. We found no difference in MIRO scores between carriers and non-carriers of *TREX1* variants (Fig. 5B; see Supplementary Fig. 6 for individual component analytes). To examine whether *TREX1* variants with a higher likelihood to result in biologically deleterious effects are associated with a proteomic interferon signature, we next classified reported disease-causing variants into high-confidence predicted loss-of-function (pLOF) vs. non-pLOF variants using the Loss-Of-Function Transcript Effect Estimator (LOFTEE)[26]. There were 30 carriers of any of 16 pLOF variants and 80 carriers of any of 19 non-pLOF variants with MIRO available (see Supplementary Table 10 for full description). Compared to non-carriers, MIRO scores were not increased in people carrying pLOF variants or non-pLOF variants (Fig. 5C). Similarly, MIRO scores were not associated with greater Combined Annotation-Dependent Depletion (CADD; v1.6)[27] scaled scores without significant interaction with LOFTEE classification (Fig. 5D). All results were comparable in 399,587 genetically unrelated participants (Supplementary Fig. 7) and were replicated in 393,433 European ancestry participants using SAIGE-GENE+[19] to account for sample relatedness (Supplementary Table 11). Likewise, no ancestry-specific trends were observed for SLE/SjD across individuals of South Asian ($n = 6429$), West African ($n = 3854$) or East Asian ($n = 1787$) ancestries, whether cases were defined from any sources or hospital records only, although sample sizes were lower (Supplementary Table 12).

## Systematic review and meta-analysis of *TREX1*-SLE genetic association studies

Narrative reviews of SLE genetics typically cite the *TREX1*-SLE association as a finding which has been replicated in further studies as well as case reports[9–13]. Given that biases can develop in narrative reviews, we performed a systematic review of genetic association studies on the

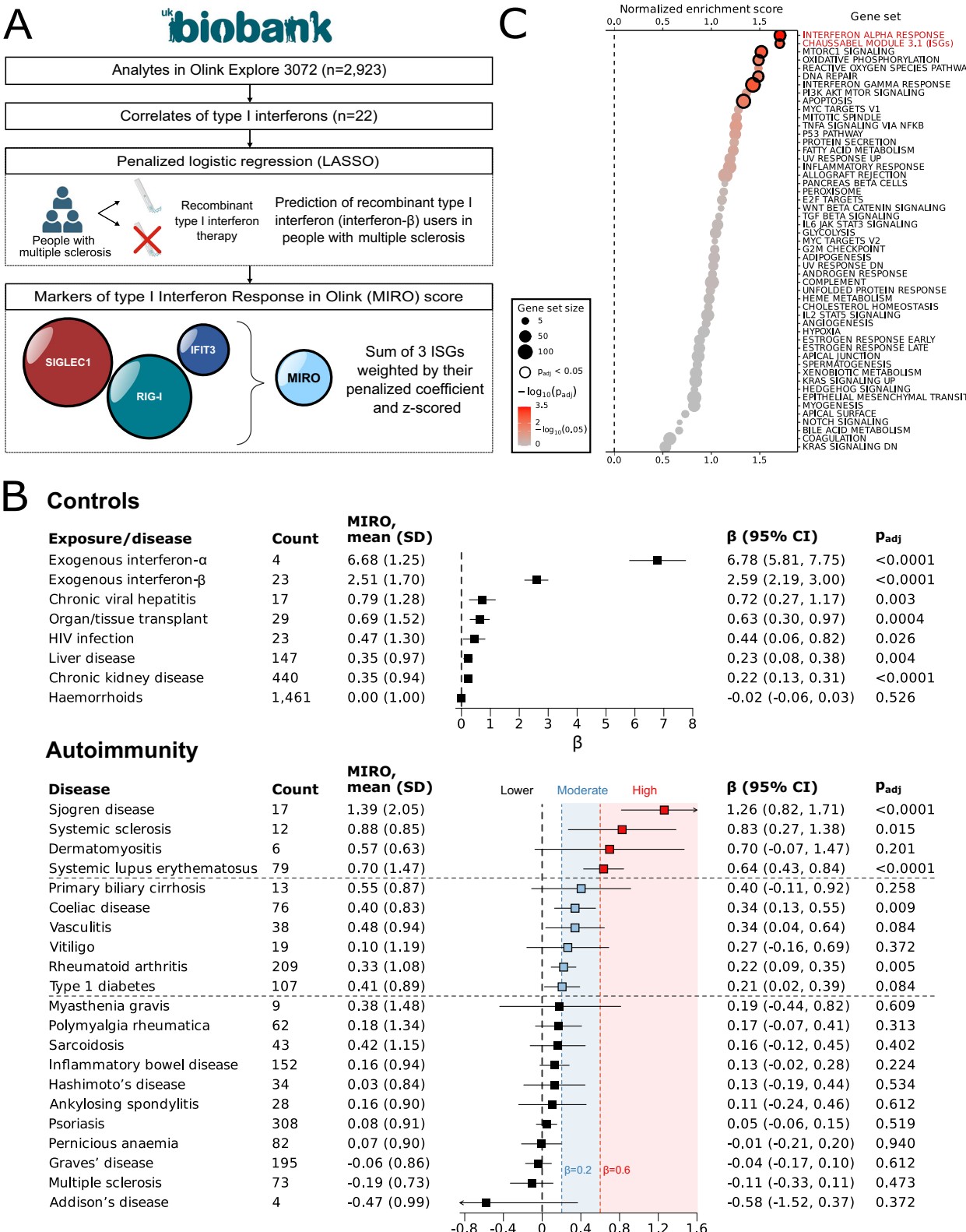

risk of SLE in carriers of rare *TREX1* variants to contrast our findings with current available evidence. We searched Ovid EMBASE and Ovid MEDLINE with concepts 'genetic variants', '*TREX1*' and 'SLE' to identify studies examining the association of rare *TREX1* variants with SLE in humans from inception to 15 August 2024 (see Supplementary Table 13 for full search strategies). References from a recent review on the genetics of SLE[28] were also added to the screening phase to ensure

studies most relevant to the field were captured. This search yielded 339 records after removing 69 duplicates, of which 23 were assessed through full-text reading to include three articles in the review (Fig. 6A). The first study (Lee-Kirsch et al.)[7] focused on the association of nonsynonymous *TREX1* variants (exon plus part of the 3'UTR) with SLE/SjD in three European cohorts (UK, Germany, Finland). The second study (Namjou et al.)[8] aimed to replicate findings from this first study

**Fig. 3 | An oligoprotein type I interferon signature derived from Olink proteomics and identification of interferonopathic autoimmune diseases in UK Biobank. A** Derivation of the MIRO score using Olink Explore 3072 analytes in UK Biobank. Circle packing of included interferon-stimulated genes (SIGLEC1, RIG-I, IFIT3) with area proportional to their absolute coefficient. Partly created in BioRender. Rioux, B. (2025) https://BioRender.com/zg7bcjo/. **B** Association of MIRO with controls and 21 common autoimmune diseases. Cross-sectional design of prevalent conditions (up to 10 years before baseline) in UK Biobank. Positive controls: interferon therapy, chronic viral hepatitis, organ/tissue transplant, HIV infection, liver and kidney disease. Negative control: haemorrhoids. Ten autoimmune conditions had a high (red) or moderate (blue) type I interferon signature (as per point estimates). Associations tested using linear regressions of MIRO (dependent variable) with each condition adjusted for potential confounders (two-sided tests). Data are presented as adjusted β (95% CI). Statistical significance defined as $p_{adj} < 0.05$ (adjusted for the number of tests within controls and autoimmunity separately). **C** Gene set enrichment analysis of MIRO across biological processes from MSigDB (hallmark) and classical ISGs. Rankings defined using Pearson's correlation coefficients of MIRO with 2906 Olink analytes and classical ISGs defined using Chaussabel module 3.1. Normalised enrichment scores and FDR-adjusted $p$ values (Benjamini–Hochberg procedure) obtained by gene set enrichment analysis. Source data are provided as a Source Data file. CI confidence interval, HIV human immunodeficiency virus, ISG interferon-stimulated gene, LASSO Least Absolute Shrinkage and Selection Operator, MIRO Markers of type I Interferon Response in Olink, MSigDB Human Molecular Signatures Database, $p_{adj}$ false discovery rate adjusted $p$ value (Benjamini–Hochberg procedure), SD standard deviation.

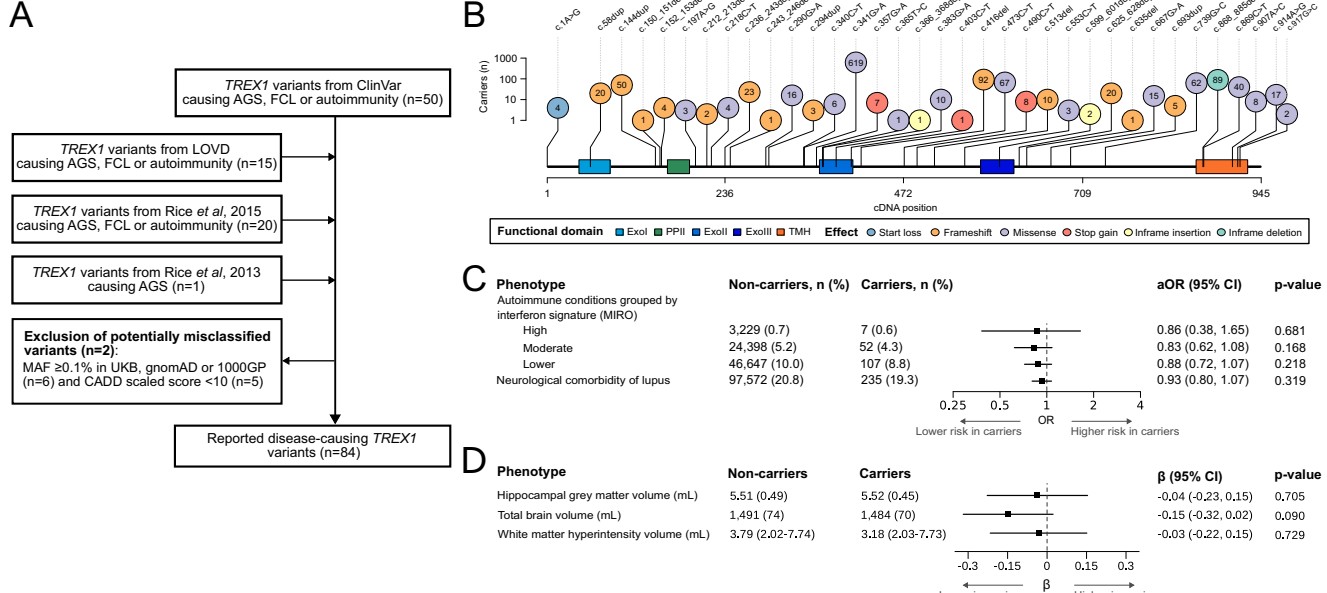

**Fig. 4 | *TREX1* variants are not associated with type I interferon-related autoimmunity in UK Biobank. A** Flow diagram for the selection of reported disease-causing *TREX1* variants. Two variants were excluded: c.679G > A and c.720G > C. **B** Lollipop plot of carrier counts for reported disease-causing *TREX1* variants in UK Biobank. Source data are provided as a Source Data file. **C** Clinical associations of reported disease-causing *TREX1* variants. Autoimmune diseases grouped by type I interferon signature from MIRO scores (Fig. 3B). Associations examined using logistic regressions with Firth penalisation adjusted for age, sex and the 10 first genetic principal components (two-sided tests). Data are presented as adjusted OR (95% CI). **D** Neuroradiological associations of reported disease-causing *TREX1* variants.

Volumes refer to mean (SD) or median (Q1-Q3). Associations examined using linear regressions adjusted for age, sex, scanning centre and the 10 first genetic principal components (two-sided tests). Data are presented as adjusted β (95% CI). 1000GP 1000 Genomes Project, AGS Aicardi-Goutières syndrome, aOR adjusted odds ratio, CI confidence interval, CADD Combined Annotation-Dependent Depletion score (PHRED-like scaled), Exo exonuclease region, FCL familial chilblain lupus, gnomAD Genome Aggregation Database exome, LOVD Leiden Open source Variation Database, MAF minor allele frequency, MIRO Markers of type I Interferon Response in Olink, PPII polyproline II motif, SD standard deviation, SLE systemic lupus erythematosus, TMH transmembrane helix, UKB UK Biobank.

and expand the association of *TREX1* variants with SLE in multi-ancestral cohorts using array genotyping for previously reported rare variants (i.e. not capturing all rare variants) and more common tag single nucleotide polymorphisms. The third study (Jiang et al.)[29] compared the frequency of rare coding variants in 76 SLE-associated genes (including *TREX1*) in small cohorts from Australia and the US. These three studies included a total of 7618 cases and 8870 controls, mostly from Namjou et al. (90.3% of cases and 75.4% of controls; see Supplementary Tables 14 and 15 for full study details and risk of bias). Lee-Kirsch et al. reported a significant association of any nonsynonymous *TREX1* variants with SLE (with a gene-level test), whereas Namjou et al. reported no significant association of individual *TREX1* variants with SLE (with variant-level tests). Jiang et al. reported the frequency of *TREX1* carriers among cases and controls without formal statistical test. When compared to the first-in-topic publication (Lee-Kirsch et al.), all subsequent studies, including UKB, had a lower prevalence of carriers in cases (0.2–1.5% vs. 1.8%) and a higher prevalence of carriers in controls (0.3–1.0% vs. 0.1%).

We meta-analysed data from these three publications with those of UKB to provide an updated pooled risk estimate. Using a random effects model, pooled estimates showed no significant association of rare *TREX1* variants with SLE (OR = 2.16; 95% CI: 0.63, 7.34; $p$ = 0.219; Fig. 6B; see Supplementary Fig. 8 for funnel plot). The substantial heterogeneity across the four studies (Q test for heterogeneity: $p$ = 0.005; proportion of total variance due to heterogeneity [$I^2$] = 76.6%) was reduced by defining subgroups based on the potential for residual confounding due to population stratification (Q test for subgroup differences: $p$ = 0.020; residual $I^2$ = 42.9%; proportion of heterogeneity explained [$R^2$] = 76.8%). None of the pooled estimates within subgroups were significant, even for studies that did not control for potential population stratification. In summary, this systematic review and meta-analysis does not support a strong role of *TREX1* variants in SLE.

Based on these findings, we next explored whether confounding by population stratification might partly explain prior positive associations. Three of the ten previously reported risk variants were >40×

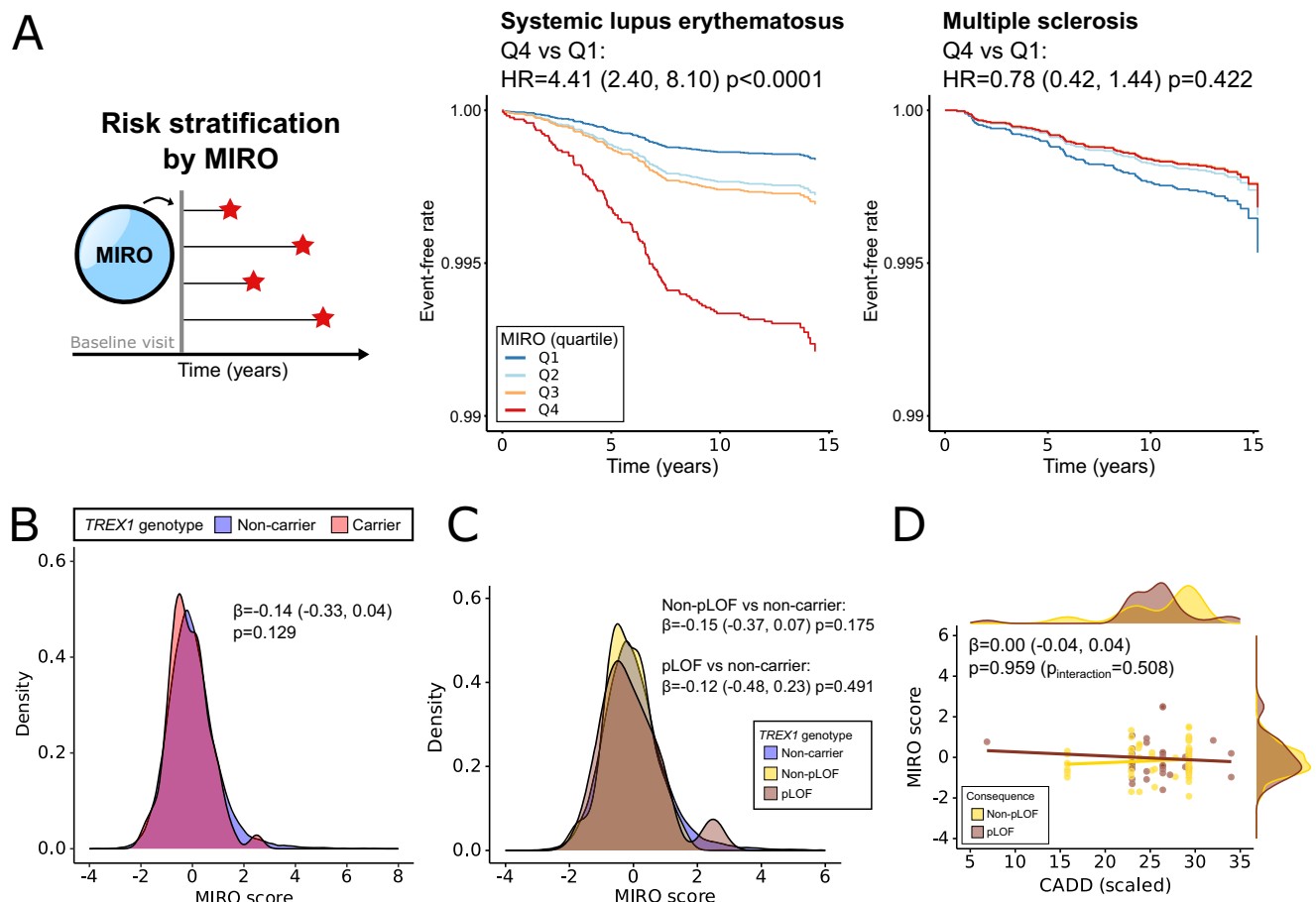

**Fig. 5 | Oligoprotein interferon signatures are associated with high risk of SLE in UKB-PPP, but are not elevated in *TREX1* mutation carriers.** **A** A peripheral proteomic signature for type I interferon activation (MIRO score) predicts incident SLE (left) but not multiple sclerosis (right). Adjusted time-to-event curves by quartile of MIRO score from Cox regression models (HRs with 95% CIs, two-sided tests). **B** Density plot of MIRO score by *TREX1* genotype. Association examined using a linear regression adjusted for age, sex and the first ten genetic principal components (two-sided test). **C** Density plot of MIRO score by LOFTEE classification. Associations examined using linear regressions adjusted for age, sex and the first ten genetic principal components (two-sided tests). **D** Scatter plot of MIRO score by CADD and LOFTEE classification. Associations examined using linear regressions adjusted for age, sex and the first ten genetic principal components (two-sided tests). CADD Combined Annotation-Dependent Depletion score (PHRED-like scaled), CI confidence interval, HR hazard ratio, pLOF high-confidence predicted loss-of-function from LOFTEE, MIRO Markers of type I Interferon Response in Olink.

more frequent in the African/African American as compared to the Non-Finnish European subpopulation of the Genome Aggregation Database (gnomAD) exome[26] (v2.1; Fig. 6C). Similarly, in UKB, the likelihood of carrying any of these 10 *TREX1* variants or being diagnosed with SLE both differed by ancestry. Specifically, non-European ancestry was more frequent in both carriers (27.0 vs. 16.1% in non-carriers) and people with SLE (24.9 vs. 16.1% without). Finally, no variants were observed in the family-based Finnish cohort controlling for population stratification and included in a previous *TREX1*-SLE study[7]. These observations suggest population stratification is a plausible source of confounding, which might have contributed to the observed heterogeneity in prior *TREX1*-SLE studies.

## Discussion

In this study, we re-examined the biologically plausible hypothesis that monoallelic variants in *TREX1* confer an increased risk for non-monogenic SLE. In UKB, rare *TREX1* variants were more common than previously reported in the general population and were not associated with SLE. We tested a wider hypothesis that rare *TREX1* variants might contribute to autoimmunity by building a novel proteomic interferon signature to identify ten type I interferon-related autoimmune conditions. In this UKB study, rare *TREX1* variants did not increase the risk of interferonopathic disease and were not associated

with endophenotypes of type I interferon autoimmunity. These neutral results are consistent with prior genetic association studies retrieved through a systematic review of the literature, which, together with UKB data, do not support the reputed strong risk of SLE conferred by rare *TREX1* variants. We finally identified population stratification as a plausible source of confounding in some earlier studies. This reappraisal of *TREX1*-SLE association studies highlights the evolving framework for interpreting genetic association studies and demonstrates the need for timely reappraisals of historical genetic association evidence.

SLE is a complex disease with strong inherited determinants, as shown by the notable difference in concordance between monozygotic (24%) and dizygotic (2%) twins[30]. Prior genome-wide association studies (GWAS) have identified susceptibility loci in genes involved in AGS (e.g. *IFIH1*) as well as type I interferon expression (e.g. *IRF5*, *IRF7*, and *IRF8*) and regulation (e.g. *SOCS1*)[31]. Rare functional variants have also been associated with SLE in genes involved in the repression of the type I interferon response, such as *BLK* and *BANK1*[29]. As such, a link between SLE and rare variants in *TREX1*, which encodes an important negative regulator of the type I interferon response, is biologically sound. Despite these priors, a recent large multi-ancestry and multi-trait GWAS meta-analysis for SLE did not identify susceptibility loci linked to *TREX1*[32], whereas our meta-analysis does not

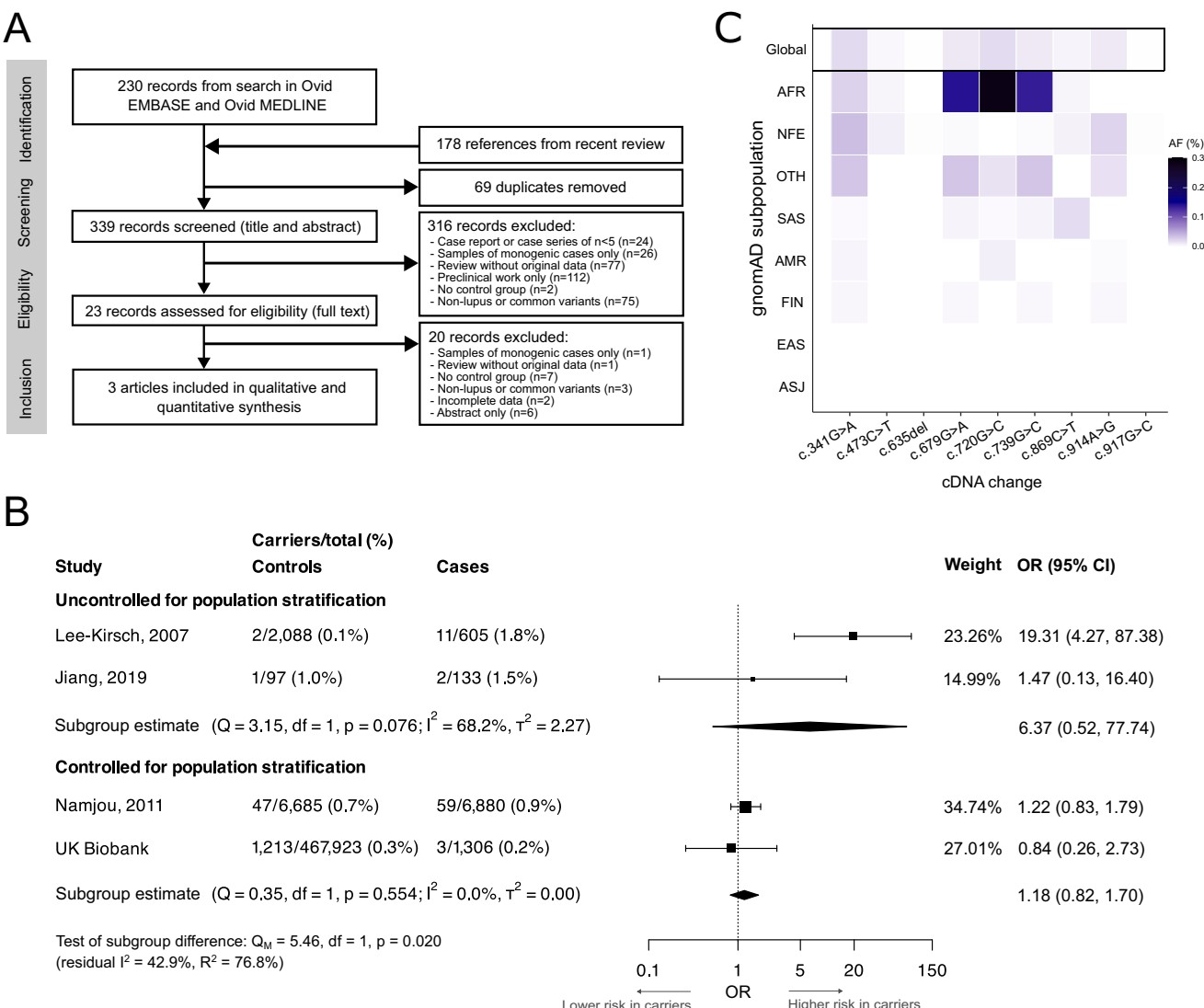

**Fig. 6 | Systematic review and meta-analysis of genetic association studies for *TREX1* variants and SLE. A** Flow diagram for study screening and inclusion in the systematic review. **B** Forest plot of previously reported *TREX1*-SLE genetic association studies meta-analysed with UK Biobank results (random effects model with subgroups based on control for population stratification). Data are presented as OR (95% CI). The overall estimate across subgroups was omitted from the forest plot due to a significant subgroup difference but was non-significant: OR = 2.16 (95% CI: 0.63, 7.34; $p = 0.219$), test of heterogeneity: $p = 0.005$ ($Q = 12.81$, df = 3), $I^2 = 76.6\%$, $\tau^2 = 1.08$, 95% prediction interval: 0.20, 23.30. Weights are presented for the overall estimate in the random effects model. Similar results were obtained with a mixed effects logistic regression model to account for sparse data (OR = 2.27; 95% CI: 0.63, 8.18; $p = 0.211$). Pooled estimates within subgroups were non-significant

(uncontrolled for population stratification: OR = 6.37, 95% CI: 0.52, 77.74, $p = 0.147$; controlled for population stratification: OR = 1.18, 95% CI: 0.82, 1.70, $p = 0.379$). For UK Biobank, carriers are defined from reported disease-causing variants and not only from reported SLE risk variants. **C** gnomAD allele frequencies for previously reported *TREX1* SLE risk variants[7]. Variant c.812_813insAA was not found in gnomAD exome and is not shown on the heatmap. Source data are provided as a Source Data file. AF allele frequency, cDNA coding DNA, CI confidence interval, gnomAD Genome Aggregation Database exome (subpopulations: AFR African/African American, AMR Latino/admixed American, ASJ Ashkenazi Jewish, EAS East Asian, FIN Finnish, NFE non-Finnish European, OTH other, SAS South Asian), OR odds ratio, SLE systemic lupus erythematosus. Source data are provided as a Source Data file.

support an association between rare *TREX1* variants and SLE. Rare variants, however, may have different clinical effects depending on their functional consequence. People with SLE, for example, do not carry substantially more rare nonsynonymous variants in SLE-associated genes as compared to people without SLE, but their rare variants are more relevant and deleterious for key immunoregulatory functions[29]. We applied this hypothesis to *TREX1*, but our findings do not suggest that variants with stronger evidence of deleteriousness confer a greater risk for SLE. In addition, only two of the 10 initially reported *TREX1* risk variants are known to cause AGS in homozygotes or compound heterozygotes (c.341G > A and c.635del)[9,20] and most of these variants (6/10) fall beyond the classical exonuclease-inactivating locations associated with AGS[3]. Similarly, our functional

characterisation of these variants suggests few of them (2/10) impair the exonuclease function of TREX1, highlighting the important role of functional validation for candidate risk variants.

*TREX1* mutations can lead to two monogenic interferonopathies, AGS[3] and FCL[5], both caused by a loss of exonuclease function leading to the accumulation of cytosolic DNA and type I interferon overstimulation[2]. Biallelic loss of function mutations underly most AGS cases, but a few heterozygous missense and frameshift mutations are reported in AGS (c.52G > C [D18H], c.583C > T [H195Y], c.598G > A [D200N], c.598G > C [D200H]), FCL (c.50T > C [F17S], c.375dup [G126Wfs], c.394C > G [P132A], c.585C > G [H195Q]) and both AGS/FCL (c.52G > A [D18N])[9]. It is well established that these monoallelic *TREX1* mutations cause monogenic immune disease, with mutations such as

c.52G > A (D18N) exerting a dominant negative effect due to its catastrophic consequences on enzymatic activity[33,34]. Within these monogenic diseases, a link with SLE is well recognised: some children with AGS manifest an early-onset form of SLE[35], while FCL is a distinct type of cutaneous lupus erythematosus. Biallelic *TREX1* mutations (c.290G > A [R97H][36] and c.341G > A [R114H][8]) have also been reported in monogenic lupus without typical features of AGS, which may be due to phenotypic variability within the clinical spectrum of *TREX1*-related monogenic disorders. Conversely, a number of case studies of individuals with SLE have reported *TREX1* mutations[37–39], often with limited functional evidence for deleteriousness. The clinical significance of such mutations is uncertain.

These results have implications for attributing disease relevance of heterozygous *TREX1* mutations found in people with SLE, because our results show that this association is weaker than previously thought. Firstly, *TREX1* variants with no functional consequence are unlikely to be disease-relevant. Secondly, monoallelic variants that do not lead to profound functional perturbation (such as seen in dominant negative nuclease-dead mutations like c.52G > A [D18N]) are unlikely to confer increased risk of SLE. This observation may also be relevant for counselling of parents of children with AGS, who have monoallelic loss-of-function *TREX1* mutations and no demonstrable increased SLE risk. Finally, there remains some very rare deleterious mutations where we do not have enough information to attribute or exclude a causal role in SLE.

The UKB-PPP offers an important opportunity to study the link between type I interferon, SLE and *TREX1* mutations. Interferon-α is typically not captured in proteomic datasets, and we show here that an oligoprotein signature can be derived from broad capture proteomics, which can act as a reasonable proxy measure of interferon-α. This allows us to leverage the additional strength of the UKB-PPP study to evaluate whether *TREX1* mutations confer risk for other 'interferonopathic' autoimmune diseases, as well as look for the presence of interferon signatures in mutation carriers. We have developed these oligoprotein interferon signatures from multiple sclerosis patients in UKB receiving recombinant type I interferon therapies, which are known to strongly upregulate interferon response genes[40]. This identifies proteins (SIGLEC1, RIG-I, and IFIT3) that are encoded by established ISGs. We have validated the MIRO score both internally and externally against interferon-α, in both patients receiving recombinant interferon-α therapy (in UKB) and in patients with high interferon-α serum concentrations (in UKPSSR). This shows that the MIRO score is a reasonable yet imperfect proxy for interferon-α concentrations in UKB. We show that interferon signatures can precede SLE diagnosis for many years, and are observed in a number of other autoimmune conditions, but such type I interferon endophenotypes are not associated with *TREX1* mutations. Given the utility of transcriptomic interferon signatures in the study of SLE, we note the potential for future refinement of this oligoprotein approach for the study of interferonopathic disorders in large, deeply phenotyped cohorts such as UKB. Further development and refinement of this oligoprotein signature approach, in particular cross-validation with transcriptomic interferon signatures across multiple disease and control cohorts, will be an important area for future study[20].

Our study has a number of limitations. First, we cannot exclude a predisposition to SLE by some *TREX1* variants due to insufficient statistical power (e.g. very rare variants or rare variants associated with disease with small effect sizes). Our UKB study, systematic review and meta-analysis use gene-level analyses and demonstrate a lack of association between *TREX1* variants and SLE. There is the potential for simple gene burden tests to dilute true signals with numerous neutral variants, which can occur despite the inclusion of diverse ancestries in the meta-analysis. It is established that very rare and highly functional deleterious loss of function mutations in *TREX1* can cause monogenic interferonopathic disease through a dominant negative mechanism (e.g. c.52G > A [D18N] and c.598G > A [D200N]), and there are other

variants implicated in atypical interferonopathic syndromes with overlap with lupus, such as c.388G > A (D130N) where disease-relevance remains unclear and cannot be addressed in this study because of rarity (there are only 6 carriers of c.388G > A in UKB, none with a diagnosis of SLE). Furthermore, non-European ancestry populations are underrepresented in UKB. The meta-analysis, however, includes a large study with SLE patients of European American (3936 cases), Asian (1265 cases), African American (1527 cases), Gullah (152 cases) and Hispanic (1492 cases) ancestries reporting neutral associations for previously reported risk variants within each group[8], suggesting that any strong ancestry-specific *TREX1*-SLE association is unlikely. These analyses highlight the importance of stringent functional assessments to examine causality in future studies of very rare *TREX1* variants that our study has limited power to dissect. Second, our analysis focused on SLE in adults and may not be generalisable to early-onset SLE, which is enriched for monogenic disease, although evidence for *TREX1* risk variants in children is limited. For example, in one study of 117 children with SLE, a single monoallelic frameshift *TREX1* variant was identified (c.236_243dup [S82Lfs]), which our analysis suggests is not associated with SLE[41]. Third, diagnosis of SLE in UKB is based on administrative healthcare records, which can be subject to various degrees of misclassification. This contrasts with more formal genetic studies, where more stringent diagnostic criteria are used. SLE diagnoses from administrative health records are prone to misclassification, leading to bias towards the null. The majority of UKB cases, however, were found through inpatient hospital records, which have a high positive predictive value (>95%)[15], and the neutral meta-analysis included a well-phenotyped cohort fulfilling the American College of Rheumatology criteria[8]. Furthermore, type I interferon signatures, which are not subject to diagnostic misclassification, were not associated with *TREX1* variants. Fourthly, UKB is a longitudinal cohort study, and while most participants who go on to develop SLE will have been diagnosed at the point of the study, it is possible that some individuals undiagnosed with SLE will develop this diagnosis in the future. Despite this limitation, the robustness of our conclusions is supported by our meta-analysis of existing genetic case-control studies. Finally, neurological manifestations of SLE are complex and heterogeneous and not limited to the six clinical phenotypes examined here. These manifestations will not be fully captured by the neurological diagnoses we have used, which is an important limitation from the perspective of studying neurolupus. Brain imaging changes are also heterogeneous in SLE, and in our neuroradiological analyses, we have analysed both brain volumes as well as white matter hyperintensity volume. The latter is a structural imaging abnormality seen in people with SLE that reflects accelerated microvascular disease[23], although it is not seen in all patients and is not a specific imaging marker for SLE. However, as yet there is no reliable neuroimaging biomarker for neurolupus, and therefore the absence of an association is not conclusive.

In summary, despite strong biological plausibility, we found no significant association between *TREX1* variants and type I interferon-associated autoimmunity, or activation of the type I interferon response in UKB. At a practical level, we suggest there is no merit in sequencing *TREX1* in patients with SLE, where monogenic disease is not suspected.

## Methods

This study complies with the Strengthening the Reporting of Genetic Association Studies (STREGA)[42] and the Preferred Reporting Items for Systematic Reviews and Meta-analyses (PRISMA)[43] statements (see Supplementary Notes 1 and 2 for checklists).

### UK Biobank

UKB is a large population-based cohort of 502k volunteers aged 40–69 years at recruitment (2006–2010) who were identified through UK

patient registries (response rate: 5.5%)[14]. Self-reported medical conditions were ascertained through questionnaires followed by verbal interviews conducted by trained staff at baseline for all and during follow-up for a subset of participants. Longitudinal health-related outcomes are captured through linkage with hospital episodes (primary or secondary diagnoses; International Classification of Diseases [ICD] v9 and v10 codes), national death registries (primary and secondary causes of death; ICD v10 codes), and primary care records (Read v2 and v3 codes; available in ~46% of the cohort). Primary care data include both diagnoses from general practitioners and those from outpatient secondary care (e.g. rheumatology clinics) as these are generally reported back to the practice[44].

We defined SLE and SjD from published ICD (v9 and v10) and Read (v2 and v3) codes, as well as UKB mappings available online (see https://biobank.ndph.ox.ac.uk/ukb/)[15,45]. We included diagnostic codes for both lupus erythematosus (ICD v10: L93.X) and SLE (ICD v10: M32.X) because of a shared pathophysiology, although the former accounted for a minority of cases (15.1% for hospital episodes used to define SLE/SjD). Expanded clinical and radiological phenotypes were defined from autoimmune diseases[46], common neurological comorbidities of SLE (due to the previous association of *TREX1* variants with neurological manifestations in SLE[8]), and radiological phenotypes observed in SLE (white matter hyperintensities[23] and brain atrophy[24]).

Brain magnetic resonance imaging (MRI) scans were obtained in a subset of the cohort on 3T Siemens Skyra scanners running VD13A SP4 with a standard Siemens 32-channel radio-frequency receiver head coil[47]. We used pre-processed imaging-derived phenotypes made available by UKB, obtained from validated segmentation algorithms (the Brain Intensity Abnormality Classification Algorithm tool [BIANCA][48] for white matter hyperintensity volume; FMRIB's Automated Segmentation Tool [FAST][49] and SIENAX[50] for total and region brain volumes). All imaging-derived phenotypes were normalised to head size using the UKB scaling factor (derived from the external surface of the skull) and *z*-scored for analyses. White matter hyperintensity volumes were log-transformed before standardising given their right-skewed (log-normal) distribution. These brain volumes were available for a range of 188-193 participants with SLE/SjD, 42,337-43,621 participants without SLE/SjD, 85-87 *TREX1* variant carriers and 42,440-43,727 non-carriers.

Proteomic analysis of blood plasma samples from ~54k UKB participants was performed by the UKB-PPP using Olink Explore 3072, a high-throughput protein biomarker discovery platform using the antibody-based proximity extension assay technology with up to 2923 unique analytes per sample[51]. Olink protein levels are quantified as normalised protein expression (NPX), a relative measure on the log2 scale with a roughly normal distribution. Readings were split into batches, which underwent within- and between-batch intensity normalisation. We used data on randomly selected (~47k) and consortium-chosen (~6k) participants at baseline, excluding those who took part in the COVID-19 repeat-imaging study (~1k). We excluded analytes with >20% missing data ($n = 12$) and NPX values were standardised for analysis (mean = 0 and SD = 1).

UKB has received ethical approval from the North West Multicentre Research Ethics Committee, and all participants provided written informed consent at recruitment. This research was conducted under UKB Resource application number 93160.

### Whole-exome sequencing and genotypes in UK Biobank

Raw sequencing data (final exome data release, July 2022) were generated with the Illumina NovaSeq 6000 platform (S2 and S4 flow cells) and a target-enrichment probe kit (IDT xGen® Exome Research Panel v1.0) to enable deep and uniform coverage of ~39 Mbp (19,396 genes). We used multi-sample project-level VCF (pVCF) files made available by UKB[52], which were obtained after (i) mapping raw sequences with the Original Quality Functionally Equivalent (OQFE) standard protocol and

(ii) producing variant calls with DeepVariant (v0.10.0; a tool that uses a deep convolutional neural network to determine the most likely genotype at each locus) and GLnexus (v1.2.6; to aggregate sample-level variant files into joint genotype pVCF files)[53]. We applied a set of quality control metrics as previously described for UKB exome[54]: (i) no withdrawal from the cohort, (ii) no mismatch between self-reported (UKB field 31) and genetically determined sex (UKB field 22001), (iii) no sex chromosome aneuploidy (UKB field 22019), (iv) individual and variant missingness <10%, (v) Hardy Weinberg equilibrium p-value $> 10^{-15}$, (vi) at least one sample per site with allele balance threshold >0.15 for single nucleotide variants (SNVs) and >0.20 for small indels, (vii) minimum read coverage depth of seven for SNVs and 10 for indels, and (viii) sequencing depth ≥10× in 90% of samples (to prevent spurious associations that may result from batch effect)[55].

### *TREX1* variants

Reported disease-causing *TREX1* variants involved in AGS, FCL or autoimmunity were retrieved from genotype-phenotype databases and publications using a systematic approach. We searched ClinVar[21] and LOVD (v3.0)[22] and included variants exclusively reported as pathogenic or likely pathogenic by individual submitters according to the American College of Medical Genetics and Genomics (ACMG) standard terminology[56]. Variants with a conflicting interpretation of pathogenicity across submitters were included if none of the individual submitters had classified variants as benign or likely benign (i.e. the conflict was between pathogenic/likely pathogenic and uncertain significance). We also included variants reported in a comprehensive review of *TREX1* genotype-phenotype associations (Rice et al.)[9] and a case-control study of 17 individuals with AGS (Rice et al.)[20]. We excluded variants fulfilling both of the following criteria to account for potential misclassification of benign variants as pathogenic: (i) MAF ≥ 0.1% in UKB, gnomAD exome (v2.1)[26], or 1000 Genomes Project[57] (including any subpopulation of these two last sequencing databases) and (ii) CADD scaled score <10, indicating a lower probability of deleteriousness. The impact of variants on the canonical transcript was assessed on Ensembl with the Variant Effect Predictor[58].

The 3D structure of TREX1 was exported from alpha-fold, which uses existing experimental structures in the Protein Data Bank and combines them with predicted structure for unresolved regions producing a per residue confidence score (AF-Q9NSU2-F1, 23/8/24). For illustrative purposes, we have shortened the long flexible linker region between the enzymatic domain and the endoplasmic reticulum transmembrane domain. SLE risk variants were mapped and highlighted in PyMoL (The PyMOL Molecular Graphics System, Version 3.0, Schrödinger, LLC). Dimer interface was approximated from previous structural studies[2].

### Functional analysis of *TREX1* variants

Gateway cloning was used to construct mammalian expression vectors. Briefly, the coding sequence of human *TREX1* (ENST00000625293.3) was amplified by polymerase chain reaction (PCR) to include attB sites and cloned into pDONR221 (Invitrogen) via BP reaction (BP Clonase II kit; Invitrogen) to generate an ENTRY vector (pENTRY-TREX1). pEGFP-TREX1 was constructed by recombining the *TREX1* coding sequence from the ENTRY vector into a Gateway converted pEGFP-C2 destination vector (Clontech) via LR reaction (LR Clonase II kit; Invitrogen). Minipreps of plasmid DNA (Qiagen) were performed for verification by Sanger sequencing (Source Bioscience). Midiprep of plasmid DNA (ZymoResearch) was performed for mammalian cell transfection. Mutations were introduced into the pEGFP-TREX1 mammalian expression construct by site-directed mutagenesis, as per the manufacturer's instructions (Q5 Site-Directed Mutagenesis Kit, NEB). Mutations were confirmed by Sanger sequencing.

TREX1 nuclease activity was assayed by transfecting *Trex1*[−/−] MEFs with pEGFP-TREX1 (WT or variants generated via site-directed

mutagenesis) using Lipofectamine 3000 (Invitrogen), as per the manufacturer's recommendations. Both ssDNA nuclease activity and TREX1 localisation are similar across the mouse and human species. Whole cell protein lysate was extracted with RIPA lysis buffer with complete mini protease inhibitor (EDTA-free, Roche), by incubation for 30 min on ice with occasional agitation. Remaining unlysed cells and debris were removed by centrifugation at $10,000 \times g$ for 10 min at 4 °C. Protein lysates were used immediately or stored at −80 °C. Protein concentration of the whole cell lysate was determined by BCA assay (Pierce).

Whole cell lysates (final concentration 100 ng/μl) were incubated with nucleic acid substrate (21-mer single stranded oligo with 3′ fluorescein and an internal DABCYL quencher (3′fl-intDABCYL-TREX1-21mer, sequence TAGACATTGCCCTCG5AGGTAC (Dabcyl dT at position marked 5, 3′ fluorescein)); final concentration 200 nM) in reaction buffer (20 mM Tris-HCl, 5 mM MgCl$_2$, 2 mM DTT, 100 μg/ml BSA) in an opaque 96-well plate. Fluorescent measurements were taken with a Clariostar platereader (483-14/530-30; BMG labtech), with temperature control set to 37° over a 90-min time course with a reading every minute. The area under the curve was calculated for each variant for each experiment. This value was then normalised against wild type (WT) for each experiment to get enzymatic activity as a percent of WT. Localisation of EGFP-TREX1 was assayed by imaging 4% PFA fixed cells grown on coverslips, at 63× on a Zeiss Axioimager. Images were analysed and assembled in Image J.

### United Kingdom primary Sjogren's syndrome registry

The UKPSSR is a national observational cohort of SjD patients fulfilling the 2002 American European Consensus Group classification criteria[59] and recruited across 30 UK centres with serum samples and whole blood (PAXgene tubes [Becton, Dickinson and Company]) stored at recruitment (Eppendorf tubes at −80 °C)[60]. RNA was assayed with the Illumina Human HT-12 v4 Expression BeadChip (Illumina). Globin signal suppression was performed using the Affymetrix Globin reduction protocol (Affymetrix). A random sample of 39 patients from UKPSSR was selected for a serum proteomic substudy quantifying 454 unique analytes with five Olink Target 96 panels (inflammation, immune response, organ damage, cardiovascular III and metabolism) and multi-subtype interferon-α levels with a high-sensitivity Simoa assay on an HD-X Analyzer as per manufacturer's instructions (Quanterix, USA). We log-transformed interferon-α levels to obtain a normal distribution for analyses and both transcriptomic and proteomic data were standardised for analysis (mean = 0, SD = 1).

Participants of the UKPSSR gave written consent for the collection of clinical and peripheral blood samples, and the research ethics approval was granted by the UK National Research Ethics Committee North West−Haydock.

### Derivation and validation of a proteomic type I interferon signature

We sought to develop an oligoprotein interferon score derived from Olink proteomics in the UKB-PPP, which we refer to as MIRO score. Because interferon-α and -β are not measured on Olink Explore 3072, we developed this score to capture their downstream proteomic response, analogous to the transcriptomic interferon score previously described for monogenic type I interferonopathies, which is defined as the median fold change in six canonical ISGs[61,62]. We first identified gene products quantified on Olink Explore 3072 that correlate with type I interferon levels from either (i) a total RNA expression fold change >20× after interferon-α or -β stimulation from a publicly available expression database on ISGs (INTERFEROME v2)[63] using in vitro experiments on human peripheral blood cells under normal conditions, or (ii) an ISG (more broadly defined using Chaussabel modules 1.2, 3.4 and 5.12[64]) with a positive association with log-interferon-α at $p$ value < 0.1 in a sample of 39 individuals with SjD from UKPSSR.

We next included these ISGs in a penalised logistic regression model using L1 regularisation (Least Absolute Shrinkage and Selection Operator or LASSO regression), which shrinks coefficients to select features and account for collinearity, to achieve the best and most interpretable prediction of type I interferon stimulation. The model was trained on UKB participants with multiple sclerosis treated vs. not treated by exogenous interferon-β therapy (any formulation). This class of injectable disease-modifying therapies is used in relapsing-remitting multiple sclerosis and induces a strong transcriptomic response of classical type I ISGs that persists for several days even in long-term users[65]. The underlying disease, multiple sclerosis, is not associated with elevated type I interferon[66], and so the interferon signatures in these patients are driven by the recombinant interferon protein. We considered multiple sclerosis diagnoses either prevalent at baseline or captured within 5 years of follow-up to balance statistical power and biological relevance, as multiple sclerosis pathogenesis precedes diagnosis by 5–10 years[67] and because event dates reaching UKB through administrative healthcare databases may lag behind actual diagnostic dates[68]. Of 235 participants with multiple sclerosis who had readings on all candidate ISGs and did not have comorbid autoimmunity, chronic kidney disease or liver disease at baseline (to limit potential noise due to other sources of inflammation), we included 21 participants on interferon-β therapy who were randomly matched (1:4 ratio) on age (5-year strata) with 84 participants who did not report using interferon-β at baseline. This sample was used to train and select the most regularised model within one standard error of the regularisation parameter, minimising the mean ten-fold cross-validated binomial deviance ($\lambda_{1se}$). The MIRO score was calculated as the sum of standardised protein levels for the three included ISGs, each weighted by their L1 penalised coefficient, followed by standardisation (mean = 0, SD = 1; see Supplementary Table 16 for weights). A score was available for 36,966 UKB participants (mean age at baseline = 56.8 years [SD = 8.2]; 54.1% females).

We validated the MIRO score in three experiments. First, we explored the cross-sectional associations at baseline of MIRO with positive control exposures (exogenous interferon-α and -β) and diseases (chronic viral hepatitis[69], organ/tissue transplant[70], human immunodeficiency virus infection[71], liver disease[72] and chronic kidney disease[73,74]) and one negative control (haemorrhoids).

Second, we performed a gene set enrichment analysis[75] using Pearson's correlation coefficients of MIRO with 2906 Olink analytes (with unique UniProt entries) as ranking metric[76]. We interrogated the 50 hallmark (H) gene sets from the Human Molecular Signatures Database (MSigDB) plus a set of classical ISGs (Chaussabel module 3.1[64]) without restricting gene set sizes (range of genes per set: 5-107). We report normalised enrichment scores (to account for gene set size in enrichment estimates) along with false discovery rate-adjusted $p$ values ($p_{adj}$) from the Benjamini–Hochberg procedure.

Third, we used a sample of 39 SjD patients with proteomics from UKPSSR to calculate Pearson's correlation coefficients for RIG-I (from Olink) and a partial MIRO score with log-interferon-α (from high-sensitivity Simoa assay) and interferon-γ (from Olink; excluding $n = 19$ with levels below the limit of detection). Because SIGLEC1 and IFIT3 were not captured in the UKPSSR Olink Target 96 panels, the partial MIRO score was calculated using CD163 as a surrogate for SIGLEC1 (correlation between the two proteins in SjD: r = 0.673, $p < 0.0001$). Pearson's correlation coefficients between log-interferon-α (from high-sensitivity Simoa assay) and expression of components of the MIRO score were also examined in 177 UKPSSR participants.

Having validated the score, we next surveyed 21 common autoimmune diseases[46] to identify type I interferon-related autoimmune conditions using MIRO. In analyses of controls and autoimmune diseases, medical conditions were defined from diagnoses up to 10 years prior to baseline after excluding people with comorbid autoimmunity, prior organ/tissue transplant (except when tested as a trait of interest)

or on interferon therapy. Associations were tested using linear regressions of MIRO (dependent variable) with each condition of interest adjusted for age (continuous) and sex. Models of medical conditions were also adjusted for interferon-γ levels (to account for its potential association with the type I interferon proteomic signature and traits) and, except for instances with low sample sizes (interferon therapy, dermatomyositis, Addison's disease and myasthenia gravis), adjusted for ethnicity (white vs. non-white), body mass index (BMI; continuous), smoking (current, previous, never), alcohol use (four intake levels: none, low, moderate, high[77]) and Townsend deprivation index[78] (continuous) at baseline.

We used Cox proportional hazards models to test the association of MIRO scores at baseline with incident diagnoses (SLE and multiple sclerosis) adjusted for potential predictors of autoimmunity (age, sex, ethnicity, smoking, Townsend deprivation index, presence of other autoimmune diseases). For each condition, participants without prevalent diagnosis were included at baseline and followed until the first of outcome occurrence, death, loss of follow-up or study end date (31/OCT/2022; last day of the month with mostly complete data for National Health Service [NHS] England inpatient records, the main source of diagnoses).

### Systematic review and meta-analysis of *TREX1*-SLE genetic association studies

We considered any study reporting a measure of association (or sufficient details on genotypes in cases and controls to build a contingency table) between rare coding sequence variants (as per individual studies but using a MAF < 1% for general guidance) in *TREX1* and diagnosis of SLE (defined as per individual studies) in humans. We did not restrict our search by genotyping/sequencing technique (e.g. Sanger or next-generation sequencing), sampling frame (e.g. specialised clinics or large biobanks) or language of publication. We did not consider studies reporting monogenic conditions which may display features of autoimmunity (e.g. AGS or FCL) as our question pertained to risk variants only. We excluded case reports and small (*n* < 5) case series as well as studies without a comparator group or without original human data (i.e. reviews, comments, editorials) and abstracts without corresponding full-text publication. This review was not pre-registered.

Two reviewers (BR and KR) independently screened records by title and abstract for eligibility, followed by full-text assessment of all records deemed eligible by any reviewer for final inclusion, at which stage disagreements were settled through consensus with a third author (DH). Reference lists of included records were explored through the same process to identify potentially missed relevant publications. Relevant abstracts were searched in Ovid MEDLINE for related full-text publications, which were included in the full-text assessment if available.

Two reviewers (BR and KR) independently abstracted relevant information using a customised spreadsheet, which included record details (authors, journal, year of publication), study samples (size, source, demographics), genotyping/sequencing technique, SLE definition, frequency of variants in cases and controls, and measures of association. Odds ratios for gene-level associations with SLE obtained after controlling for population stratification were preferentially used. If not reported, crude ORs were considered (either abstracted or calculated from reported frequencies with continuity correction), and other measures of association were converted to ORs. Discordant data were jointly re-abstracted, and any conflicting interpretation was settled through consensus with a third author (DH). Missing data relevant to the review were queried by emailing contact authors. Risk of bias was assessed with a validated tool for genetic association studies (Q-Genie) to report 11 domain-specific ratings (averaged between the two reviewers) and a summary quality assessment (as per Q-Genie guidance)[79].

Key features of the included studies were first synthesised qualitatively. Meta-analyses were then performed to pool *TREX1*-SLE associations across included studies in addition to UKB data to provide updated results. UKB estimate and variance were obtained from a logistic regression adjusted for age, sex and the first ten genetic principal components (PCs). Meta-analyses were first conducted using a random effects model with the DerSimonian and Laird method and were replicated with a mixed effects logistic regression model to account for potential bias due to sparse event data[80]. We report summary measures along with 95% CIs (to quantify the accuracy of the mean) and 95% prediction intervals for the overall estimate (to quantify the dispersion of true effect sizes about the mean). Heterogeneity was explored visually with forest plots and statistically with $\tau^2$, the $I^2$ statistics and Q tests. We performed subgroup analyses by potential for residual confounding due to population stratification to explore sources of heterogeneity. The funnel plot is presented, but statistical tests for publication bias were not performed because there were too few studies to yield sufficient power.

### Genetic analyses

The association between *TREX1* genotype and binary outcomes was tested using logistic regressions adjusted for age, sex and the first ten genetic PCs (UKB field 22009) to account for population stratification[81]. Firth penalisation was used to reduce bias in maximum likelihood estimators due to data sparsity, and point estimates are presented along with profile penalised log likelihood confidence intervals[82]. Association tests for imaging-derived phenotypes were performed using linear regressions adjusted for the same covariates, in addition to scanning centre (to control for potential technical confounding)[83]. Proteomic changes were compared by SLE/SjD status (with vs. without) and *TREX1* genotype (carriers vs. non-carriers) using linear regressions adjusted for age and sex (and the first ten genetic PCs for *TREX1* genotype).

Analyses were replicated after restriction of the cohort to participants without close relatives up to the third degree of relatedness (from kinship coefficients made available by UKB[81]) using GreedyRelated (v1.2.1; https://gitlab.com/choishingwan/GreedyRelated) to preferentially retain carriers (*n* = 69,642 excluded). Ethnic background was defined from self-reports (UKB field 21000), and European ancestry was defined from PCs in participants with self-reported 'White British' background (UKB field 22006). Non-European ancestry groups were based on a published approach using Euclidean distances on the PC space[84] and genome-wide genotyping data (UKB field 22009)[81]. The association of *TREX1* variants with clinical outcomes and MIRO scores was also replicated for participants of European ancestry (*n* = 393,433) with burden tests (for effect estimates) and optimal sequence kernel association tests (SKAT-O; for *p* values) implemented in SAIGE-GENE+ to account for sample relatedness (using logistic/linear mixed models from a sparse genetic relationship matrix adjusted for age, sex and the first ten PCs in addition to scanning centre for radiological traits) and to control for type I error inflation due to case-control imbalances and data sparsity from ultra-rare variants[19]. We defined nominal significance as *p* value < 0.05, and used a more stringent statistical significance threshold to account for multiple testing across outcomes for the main genetic analysis (*p* value < 0.0010, i.e. 0.05/49 tests; clinical: *n* = 35, radiological: *n* = 3, proteomics: *n* = 11).

Analyses were performed on the UKB Research Analysis Platform using JupyterLab (v3.6.1) with Hail (v0.2.78), Python (v3.9.16) and R (v4.2.0). Logistic and Cox regressions were conducted on R using logistf (v1.26.0) and survival (v3.7.0), and meta-analyses were conducted using metafor (v4.6-0). Visualisation of genomic and phenotype data was conducted on R using trackViewer[85], UpSetR (v1.4.0), forestploter (v1.1.1) and ggplot2 (v.3.4.4).

### Reporting summary

Further information on research design is available in the Nature Portfolio Reporting Summary linked to this article.

## Data availability

All data are included in the Supplementary Information or available from the authors, as are unique reagents used in this Article. The raw numbers for charts and graphs are available in the Source Data file whenever possible. UK Biobank data supporting the findings can be accessed through application to the resource: https://www.ukbiobank.ac.uk/enable-your-research/apply-for-access. Source data are provided with this paper.

## Code availability

Code used to generate *TREX1* association studies is available at: https://github.com/barioux/UKB_TREX1.

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

## Acknowledgements

B.R. is supported by the Centre for Clinical Brain Sciences of the University of Edinburgh (Rowling & Dr Hugh S P Binnie scholarship) and the Canadian Institutes of Health Research (CIHR; Doctoral Foreign Study Award, DFD-187711). S.M. is supported by the Clayco Foundation for RVCL research. W.W. is supported by the Chief Scientist Office of the Scottish Government (CAF/17/01), the UK Alzheimer's Society and the Stroke Association, the National Institute for Health and Care Research

(NIHR) and the National Institutes of Health (NIH). D.P.J.H. is supported by a Wellcome Trust Senior Research Fellowship (215621/Z/19/Z), the Medical Research Foundation, the UK Dementia Research Institute UK-DRIEdin0011 (Principal funder the Medical Research Council) and the Chief Scientist Office (Scottish Government, Precision Medicine Alliance, PMAS/21/06), and the UK Medical Research Council/Medical Research Foundation Emerging Leaders Prize in Lupus. We thank Professor Yanick Crow for critical reading and advice on the manuscript, and thank Kyle Thompson for help with transcriptomic analyses.

## Author contributions

All authors substantially contributed to this work and approved the manuscript. B.R., W.W. and D.H. were responsible for the conceptualisation of the study. B.R. was responsible for the methodology and analyses in UKB. S.M., D.F., K.R. and A.K. were responsible for the methodology and analyses of functional studies. J.B. and W.F.N. contributed to data curation and analyses in UKPSSR. N.D. contributed to the interpretation of results.

## Competing interests

W.F.N. has undertaken clinical trials and provided consultancy or expert advice in the area of Sjogren disease to the following companies: GlaxoSmithKline, MedImmune, UCB, AbbVie, Takeda, Resolves Therapeutics, Sanofi, Novartis, Bain Capital and Argenx. The other authors declare no competing interests.
