## [Transparent Peer Review File · Nature Communications]

Oligoprotein type I interferon signatures, but not TREX1 variants, increase risk of systemic lupus erythematosus in UK Biobank

Corresponding Author: Professor David Hunt

Version 1:

Reviewer comments:

Reviewer #1

(Remarks to the Author)

The study re-evaluated the association between TREX1 variants and lupus using exome sequences data from UK Biobank participants, alongside other methodologies. The researchers reported no association between reported TREX1 variants and non-monogenic lupus, type I interferon-associated autoimmune diseases, or interferon signatures in blood. These findings are intriguing, given that TREX1 variants are widely believed to increase the risk of lupus. The study also suggested that the previously reported associations between lupus and TREX1 risk variants might have been confounded by population stratification. I have the following comments:

1. The researchers did not replicate the previously reported TREX1 gene burden association with lupus, type I interferon-associated autoimmune diseases, or interferon signatures in the UK Biobank. The replication study appears robust, as multiple methods yielded consistent results. However, it is still challenging to conclude definitively that there is no association between TREX1 variants and SLE, type I interferon signatures, or autoimmunity. It is possible that only very rare mutations are causal in this gene. If this hypothesis is true, there may be insufficient power to detect such effects due to the extreme rarity of these variants, or the combined gene burden might be primarily driven by non-causal variants with relatively higher minor allele frequencies (MAF). In fact, the authors observed that only the two frameshift variants (P212Hfs and D272Rfs) affected protein localization, and both variants had very low MAF. Consequently, no significant association between reported TREX1 variants and lupus was observed. The authors should discuss these points in greater detail, as some of the conclusions drawn from the data may be overstated and should be moderated.

2. Evidence from murine models regarding some TREX1 mutations, such as D18N and P61Q, has shown an impact on type I interferon-driven signatures and autoimmunity, although the MAF of these mutations remains very low. This is also supported by a recent study involving 55 patients with SLE, which reported that one 27-year-old SLE patient with overlapping dermatomyositis carried a heterozygous p.Asp130Asn mutation in the TREX1 gene. This patient had significantly elevated serum interferon (IFN)- α levels compared with healthy controls and other patients with SLE. Therefore, I suggest that the following pertinent reports should be discussed to provide a more comprehensive view of the evidence:

DOI: 10.1016/j.clim.2021.108732

DOI: 10.1093/hmg/ddae089

DOI: 10.4049/jimmunol.1900220

DOI: 10.1016/j.jaut.2019.03.001

DOI: 10.4049/jimmunol.1600722

DOI: 10.1073/pnas.2411747121

3. The age distribution at diagnosis of SLE patients in the UK Biobank should be detailed. For instance, early disease onset (e.g. 20-year-old) in lupus, which is associated with higher serum IFN α activity (DOI: 10.1002/art.23619). However, since UK Biobank participants were aged 40-69 years at enrollment during 2006-2010, it remains unclear whether those with early disease onset represent a small proportion of this study. This uncertainty may affect the gene burden estimation for TREX1.

(Remarks on code availability)

Reviewer #2

(Remarks to the Author)

Re-evaluating the association between TREX1 variants, systemic lupus erythematosus and 1 oligoprotein interferon signatures in UK Biobank

Overall, this is an interesting manuscript but will require major editing before it is suitable for publication. The genetic association of rare variants in the smaller study by Lee et al did not replicate in the larger study is the strength of this paper. Most of the variants in Lee et al did not have any functional validation, except for the 2 fs mutations which were found to affect localization, replicated in this work. I think that in the current age of large genomic analysis this paper should conclude that functional validation of novel rare variant associations is necessary due to the issues of population stratification. I think the meta-analysis is also strong.

The IFN scores and the MRI associations have many methodological drawbacks that either need to be very clearly stated and included in the limitations or removed.

The genetic ancestry of the UK Biobank is largely European and may add another degree of important bias that should be addressed in limitations.

We first show that TREX1 variants 21 do not confer risk for lupus in UK Biobank and that most reported risk variants are functionally neutral 22 in mutagenesis experiments.

Moreover, UKB participants were aged 40-69 years at 62 enrolment in 2006-2010, meaning that the large majority of participants predisposed to SLE will by 63 now have been diagnosed as SLE incidence peaks in middle age.

This is true, but SLE can be diagnosed after enrolment, and this won't be detected in the database. I think the cross-sectional nature of the study is a major limitation that must be mentioned.

the genetic association of rare variants in the smaller study by Lee et al did not replicate in the larger study is the strength of this paper. Most of the variants in Lee et al did not have any functional validation, except for the 2 fs mutations which were found to affect localization, replicated in this work. I think that in the current age of large genomic analysis this paper should conclude that functional validation of novel rare variant associations are necessary due to the issues of population stratification?

~469k participants and reliable capture of SLE diagnoses 61 through routinely collected healthcare data. ~54k UKB participants, also offers an opportunity to develop and detect 66 oligoprotein interferon signatures which form an important part of the immune endophenotypes 67 associated with SLE.

What is the demographic constitution of the 54k which have proteomic data. Is there a difference in the demographics of this cohort compared to those on whom we have exome data? It is only 10% of the sample. There is a large potential for bias here.

We use this oligoprotein signature 25 to identify nine autoimmune diseases associated with elevated type I interferon: primary biliary 26 cirrhosis, dermatomyositis, lupus, Sjogren disease, systemic sclerosis, Addison's disease, type 1 27 diabetes, coeliac disease and rheumatoid arthritis. Using this approach, we find no association between 28 TREX1 variants, type I interferon-associated autoimmune disease nor interferon signatures in blood.

systematic review and meta-analysis of published studies confirms the lack of support for the 30 association between lupus and TREX1 risk variants and suggests a potential for strong associations to 31 be confounded by population stratification

10 coding TREX1 risk variants initially reported⁷ are not associated with SLE 71 or SjD in UKB.

I am curious to why the authors only selected the variants in this study, which is an old association study without much functional validation.

We then develop a proteomic interferon signature to define a cluster of type I 72 interferon-related autoimmune conditions. We show that TREX1 variants causing monogenic disease 73 are not associated with this proteomic interferon signature and do not confer risk for such 74 interferonopathic autoimmunity, even after in silico functional annotations to enrich variants for 75 deleteriousness.

Page 5, line 92- Please do use the term white for race or European for ancestry and do not use Caucasian: Popejoy AB. Too many scientists still say Caucasian. Nature. 2021;596(7873):463-.

We therefore sought to develop and validate a proteomic signature 129 of type I interferon activation. We defined 12 Markers of Interferon-_γ Response in Olink (MIROs) 130 from analytes on the Olink Explore 3072 platform and built a composite oligoprotein measurement (the 131 MIRO score) which averages their standardized levels (Figure 3A).

Did the authors do a training and test set for the MIRO validation? Using a new, not validated measure of Type I IFNs without more detailed validation gives me pause. The proteins selected are IFNs but very few are part of the validated scores most commonly used for SLE. I do not see enough validation to include this in the paper.

Kim H, de Jesus AA, Brooks SR, Liu Y, Huang Y, VanTries R, Montealegre Sanchez GA, Rotman Y, Gadina M, Goldbach-Mansky R. Development of a validated interferon score using NanoString technology. *Journal of Interferon & Cytokine Research*. 2018 Apr 1;38(4):171-85.

Rice GI, Melki I, Frémond ML, Briggs TA, Rodero MP, Kitabayashi N, Oojageer A, Bader-Meunier B, Belot A, Bodemer C, Quartier P. Assessment of type I interferon signaling in pediatric inflammatory disease. *Journal of clinical immunology*. 2017 Feb;37:123-32.

Page 18, lines 360-362:

These brain 360 volumes were available for a range of 188-193 participants with SLE/SjD, 42,337-43,621 participants 361 without SLE/SjD, 85-87 TREX1 variant carriers and 42,440-43,727 non-carriers.

Again, this is a small subset of the entire cohort. Was an analysis done to determine if they differ from the larger cohorts in important ways which may bias the results?

Page 19 lines 388-392- the exclusion criteria seem to be too liberal. Why did they choose a CADD <10, when 12-20 is the current cutoff, and with a population based approach, searching for rare, deleterious variants seems most pragmatic?

The section of the paper examining TREX 1 variants with interferonopathic autoimmunity seems to be a different story. If the paper wants to look at all causes of Type I IFNopathies, they should assess it, but including these here seems to dilute the paper. Again, the choice of variants to assess may not be stringent enough and bias the results to the see no association.

Page 22, line 458: We replicated these neutral associations with RF and CRP, two blood biomarkers associated 458 with lupus.

These are not common lupus biomarkers. ANA and ESR are much more commonly associated with lupus. Overall, I think adding someone with rheumatology expertise and specifically lupus expertise to the study team would greatly enhance the manuscript.

The GitHub link provided did not work for me to review the code.

(Remarks on code availability)

The GitHub link provided did not work for me to review the code.

Reviewer #3

(Remarks to the Author)

The authors have taken a revisionist approach using unbiased methods to the question of the associations of TREX gene variants with non-monogenic systemic lupus erythematosus (SLE). Using the exome sequencing data from the large UK Biobank linked to health system diagnostic codes, and further using a protein signature claimed to represent type I interferon (IFN) activation, they find no evidence for an association of TREX variants with SLE or IFN-associated proteins and conclude with a clinical recommendation against TREX sequencing in the management or diagnosis of SLE.

On face value these are negative studies of value to the research community. They underpin the disconnect between monogenic interferonopathies such as AGS where TEX mutations are clearly causal, and the syndrome diagnosed as SLE in adults, notwithstanding the overlap between these conditions and what the former has taught us about the latter.

The paper lacks a section on its limitations, which should be added, along with addressing other matters listed below.

1. Medical record diagnosis is notoriously weak, in particular in relation to heterogeneous autoimmune diseases where both under- and over-diagnosis are common. 35% in total of cases had the diagnosis based on a primary care-assigned diagnosis, which in this reviewer's experience is likely to represent a significant proportion of false positive SLE diagnoses, weakening any associations.

2. Ancestry linkage studies are informative, suggesting a possible reason for previous reported associations between TREX variants and SLE. However, UKB is hugely dominated by white/European ancestry subjects, known to have the lowest prevalence and severity of SLE. Significant missing ancestries include East Asian ancestries, in which with a very high prevalence of young onset SLE may have a different result. African ancestries are also <>10% of the cohort.

3. The cohort is also light-on for young onset disease, where genetic associations are believed to be stronger. <2% of cases were diagnosed at age <20.

4. The so-called MIRO protein signature is entirely derived from public domain sources, namely mRNA expression data after IFN-alpha stimulation, and predicted secretion of the gene products. This is a far from ideal way to develop such a signature, which ideally would be based on detection of proteins from human cells exposed to IFNAR activation, which in turn would not be limited to IFN-alpha stimulation as multiple other type I IFNs are implicated in SLE. Earlier work such as that of Bauer et al (10.1371/journal.pmed.0030491.st002) combined measurement of proteins directly in patient samples with in vitro exposure of cells to IFN in order to increase confidence that the signature was valid; that has not been done here.

Experimental replication of this signature is required, and its lack is a significant caveat.

5. Similarly, the 'shoulders' of individual MIRO protein levels greater than those in the health population are modest (Suppl Figure 4), notwithstanding the small p values for the difference in means potentially driven by the large sample size, though somewhat more impressive for the composite MIRO score.

6. The meaning of Figures 3 C&D is unclear
7. Figure 4C appears to be monochromatic but the legend refers to red and black components.
8. It is not immediately clear what data are represented in Figure 5B – I take it the dots, box, and whiskers are individual MIRO proteins? Each is counted when >2SD - above what?
9. Noting that IFNs are generally low abundance in human serum samples, they are detectable in some individuals. Were any type I IFN proteins detected in any sample? Did they relate to TREX variants?
10. CRP is an unusual biomarker to choose in the setting of SLE, as in most studies it has little or no correlation with disease activity and is commonly measured in the reference range in people with SLE, even when active. I do not recognise this as a valid reference biomarker for the research question being addressed here and recommend deletion of this section.
11. In vitro variant testing was done in murine cells – does this risk missing functionally important effects restricted to human cells?
12. The authors appear to be from a neurosciences centre, but apart from this why were CNS manifestations of SLE chosen to be called out?
13. The discussion is too long. TREX sequencing is not part of the diagnosis or clinical care of SLE.

(Remarks on code availability)

Version 2:

Reviewer comments:

Reviewer #1

(Remarks to the Author)

The authors' response has reduced some of my concerns, but several points warrant further discussion. The UKB and meta-analyses remain based on gene-level burden analyses, where simple gene burden tests may dilute true signals with numerous neutral variants, despite inclusion of diverse ethnicities (predominantly European). Moreover, genuine pathogenic variants such as p.Asp130Asn may not have extremely high ORs but could still contribute to sporadic SLE with elevated serum IFN α (DOI: 10.1016/j.clim.2021.108732). These are missense mutations rather than established dominant-negative mutations like D18N and D200N. The discussion of these critical points seems to be insufficient, with the authors instead vaguely claiming that the meta-analysis has adequate power without addressing the fundamental methodological limitations of burden testing in detecting variant-specific effects.

(Remarks on code availability)

Reviewer #2

(Remarks to the Author)

The authors have revised the manuscript, which I still believe has merit and has findings that contribute to the body of literature on the role of TREX1 variants in SLE.

However, I find that there continue to be methodological issues which have not been fully addressed.

I think the authors need to clarify that heterozygous TREX1 variants are not equal, and while some contribute to SLE others do not and finding a rare variant in TREX1 requires functional evidence before considering causality. This should be reflected in the title.

I do not think the authors have sufficiently addressed the weakness of SLE diagnosis from medical record as a limit

The meta-analysis is based on 2 very old studies with 2 very different approaches to sequencing (one is an Illumina chip assay, which is used for the large, diverse cohort, and may miss many rare novel variants in TREX1. This assay would be the weakest to detect variants in non-europeans populations) This needs to be stated as a limitation.

The Miro validation is not sufficient. The use of a cohort of 39 patients as a validation cohort is not sufficient. Also, to use a partial MIRO score which substitutes CD163 for SIGLEC1 is not robust methodology. The authors claim that many of the markers used are similar to validated scores does not substantiate the use of an unvalidated score. Also, the p value and r score for RIG-I are marginal in this validation. They either need to do a large validation of the method or drop this section of the manuscript.

Additionally, adding in MRI features often found in SLE does not address the limitations. There are no official MRI biomarkers for CNS SLE, so the lack of findings is not conclusive.

Finally the point that some of this cohort may be diagnosed with SLE after the bio sample was entered was not mentioned in the study.

(Remarks on code availability)

Reviewer #3

(Remarks to the Author)

The authors have responded appropriately to the comments raised in my review and the revisions are acceptable from the point of view specifically of my questions.

(Remarks on code availability)

Version 3:

Reviewer comments:

Reviewer #1

(Remarks to the Author)

The revisions address my concerns and the manuscript is now acceptable.

(Remarks on code availability)

Reviewer #2

(Remarks to the Author)

The authors have revised the manuscript, which I still believe has merit and has findings that contribute to the body of literature on the role of TREX1 variants in SLE. However, I find that there continue to be issues with clear limitations to the manuscript which have not been fully addressed.

The language used, both in the title and throughout the paper, overstates the results with the limited data and cross-sectional design of the study. The SLE diagnosis from large data sets is not simply subject to misclassification bias. The cross-sectional design is the largest limitation here. We have no way of knowing that the subject without an ICD code for SLE at the time of the data pull did not go on to develop SLE 6 months or a year after. To state that most patients are in their 40-60s and most lupus patients present before that not sufficient. Patients present at a range of ages and we cannot make assumptions based on missing data. I would remove that statement that most patients will have been diagnosed, and simply state that you have cross sectional samples and the lack of prospective data for risk prediction is a limitation.

Page 9 line 174: oligoprotein is misspelled.

I agree with reviewer 1 on the dilution of variants in burden testing.

I thank the reviewers for the detailed review of the MIRO testing. I think that validating a set of protein biomarkers for the first time in a manuscript requires a lot of effort with a high burden of proof, as they want to be using a validated test to make a conclusion. The authors have done a lot to provide evidence that MIRO is measuring Type I IFN.

I think the neurologic association study also remains a weak portion of the paper.

I recommend the authors remove this section, as it there are no validated MRI biomarkers in SLE, and more advanced MRI technologies are correlated with SLE CNS findings.

An ICD code for 6 common associations in SLE can miss many other features of neuroSLE. The authors should state in the limitations that there are no official MRI biomarkers for CNS SLE, so the lack of findings is not conclusive. Conventional MRI is not very sensitive for SLE brain changes. Conventional structural MRI abnormalities, such as white matter (WM) hyperintensities, gross brain tissue atrophy and ventricular enlargement in response to this atrophy are not always observed in patients with NPSLE. In contrast, more advanced postprocessing techniques enable the quantification of structural brain metrics beyond total tissue volumes from standard T1-weighted MRI, such as regional volumes, surface area and cortical grey matter (GM) thickness. Furthermore, other modalities such as diffusion MRI, have become more useful current NPSLE clinical research. Altered tissue microstructure has been reported in several brain regions of patients with NPSLE diagnosis when compared with healthy controls and they have correlated with higher cognitive dysfunction.

If these limitations can be clearly stated in the manuscript, I believe it is suitable for publication.

(Remarks on code availability)

Oligoprotein type I interferon signatures, but not *TREX1* variants, increase risk of systemic lupus erythematosus in UK Biobank

Many thanks for the encouraging response to our original submission. We are pleased that the reviewers felt our manuscript was of interest. Please find below a point-by-point rebuttal to the constructive comments of the reviewers, and a fully revised manuscript.

In particular we have given careful consideration to the concerns regarding the proteomic interferon signature score. We have fully addressed these, further developed the rigour of the methodology and provided additional validation.

We hope the reviewers find that we have addressed their comments.

Colour code

Black = reviewer comments

Purple = author response

Green = new text in revised manuscript

Reviewer 1

Comment 1: The researchers did not replicate the previously reported *TREX1* gene burden association with lupus, type I interferon-associated autoimmune diseases, or interferon signatures in the UK Biobank. The replication study appears robust, as multiple methods yielded consistent results. However, it is still challenging to conclude definitively that there is no association between *TREX1* variants and SLE, type I interferon signatures, or autoimmunity. It is possible that only very rare mutations are causal in this gene. If this hypothesis is true, there may be insufficient power to detect such effects due to the extreme rarity of these variants, or the combined gene burden might be primarily driven by non-causal variants with relatively higher minor allele frequencies (MAF). In fact, the authors observed that only the two frameshift variants (P212Hfs and D272Rfs) affected protein localization, and both variants had very low MAF. Consequently, no significant association between reported *TREX1* variants and lupus was observed. The authors should discuss these points in greater detail, as some of the conclusions drawn from the data may be overstated and should be moderated.

We thank the reviewer for this comment. We would highlight first that the first study (Lee-Kirsch et al. Nature Genetics 2007) to demonstrate a *TREX1*-SLE association found a strongly increased relative risk of ~25. Our UK Biobank study is adequately powered to test this widely-cited strong risk conferred by rare variants, and our metaanalysis approach further strengthens this.

We agree with the reviewer that it may be very difficult to disprove the hypothesis that only ultra rare mutations predispose to the development of sporadic lupus. Our combined UKB analyses, taken together with our meta-analysis do examine rare variants and together offer considerable power, and would suggest this is not the case.

Principal funder:

Medical
Research
Council

We would also highlight that it is clear that mutations in *TREX1* can cause monogenic interferonopathic autoimmunity (e.g. biallelic loss of function mutations or dominant negative mutations cause AGS and monoallelic mutations can cause familial chilblain lupus). Some of these monogenic disorders can display elements of a lupus-like phenotype, although they are a distinct entity from SLE.

We agree that we need to make these points clearer, and have modified the discussion to add this important nuance and moderated some statements to avoid overinterpreting our results. In particular we have now added substantial section at the end of the discussion on the limitations of our study and these include the following modifications:

Discussion (last paragraph on limitations): “Our study has a number of limitations. First, we cannot exclude a predisposition to SLE by some very rare *TREX1* variants due to insufficient statistical power, especially in non-European ancestry populations which are underrepresented in UKB. The meta-analysis, however, includes a large study with SLE patients of European American (3,936 cases), Asian (1,265 cases), African American (1,527 cases), Gullah (152 cases) and Hispanic (1,492 cases) ancestries reporting neutral associations within each group⁸, suggesting that any strong ancestry-specific *TREX1*-SLE association is unlikely.”

Discussion (1st paragraph): “In this UK Biobank study, rare *TREX1* variants did not increase the risk of interferonopathic disease and were not associated with endophenotypes of type I interferon autoimmunity.”

Discussion (on overlap between monogenic interferonopathy and SLE – see response to comment 2 below)

Comment 2: Evidence from murine models regarding some *TREX1* mutations, such as D18N and P61Q, has shown an impact on type I interferon-driven signatures and autoimmunity, although the MAF of these mutations remains very low. This is also supported by a recent study involving 55 patients with SLE, which reported that one 27-year-old SLE patient with overlapping dermatomyositis carried a heterozygous p.Asp130Asn mutation in the *TREX1* gene. This patient had significantly elevated serum interferon (IFN)- α levels compared with healthy controls and other patients with SLE. Therefore, I suggest that the following pertinent reports should be discussed to provide a more comprehensive view of the evidence:

DOI: 10.1016/j.clim.2021.108732

DOI: 10.1093/hmg/ddae089

DOI: 10.4049/jimmunol.1900220

DOI: 10.1016/j.jaut.2019.03.001

DOI: 10.4049/jimmunol.1600722

DOI: 10.1073/pnas.2411747121

We thank the reviewer for these comments. We agree that there is a need to address case reports, and we have included those suggested. Some of these, such as D18N and D200N are established dominant negative mutations (hence cause monogenic disease causing a distinct autoinflammatory syndrome with lupus-like features, rather than “sporadic” SLE). However others may well simply represent false positive associations, which can falsely amplify the

appearance of a gene association without adding any statistical weight. Single cases do not add weight of evidence if the underpinning genetic association is not sound.

Discussion: “It is well established that these monoallelic *TREX1* mutations cause monogenic immune disease, with mutations such as c.52G>A (D18N) exerting a dominant negative effect due to its catastrophic consequences on enzymatic activity. Within these monogenic diseases, a link with lupus is well recognized: some children with AGS manifest an early-onset form of SLE while FCL is a distinct type of cutaneous lupus erythematosus. Biallelic *TREX1* mutations (c.290G>A [R97H] and c.341G>A [R114H]) have also been reported in monogenic lupus without typical features of AGS, which may be due to phenotypic variability within the clinical spectrum of *TREX1*-related monogenic disorders. Conversely, a number of case studies of individuals with SLE have reported *TREX1* mutations, often with limited functional evidence for deleteriousness. The clinical significance of such mutations is uncertain, but our results suggests that the SLE risk conferred by such mutations is not as strong as previously thought.”

Comment 3: The age distribution at diagnosis of SLE patients in the UK Biobank should be detailed. For instance, early disease onset (e.g. 20-year-old) in lupus, which is associated with higher serum IFN α activity (DOI: 10.1002/art.23619). However, since UK Biobank participants were aged 40-69 years at enrollment during 2006-2010, it remains unclear whether those with early disease onset represent a small proportion of this study. This uncertainty may affect the gene burden estimation for *TREX1*.

We thank the reviewer for this comment and we have made the age breakdown for lupus patients in UKB more prominent in the text (originally Supplementary Table 3). We have also added a limitation on the few early-onset cases in UKB.

Introduction: “Moreover, UKB participants were aged 40-69 years at enrolment in 2006-2010 and the study captures both incident and prevalent diagnoses, meaning that the large majority of participants predisposed to SLE will by now have been diagnosed, as SLE incidence peaks in middle age.”

Discussion (last paragraph on limitations): “Second, our analysis focused on SLE in adults and may not be generalizable to early-onset SLE which is enriched for monogenic causes, although evidence for *TREX1* risk variants in children is limited. For example, in one study of 117 children with SLE, a single monoallelic frameshift *TREX1* variant was identified (c.236_243dup [S82Lfs]), which our analysis suggests is not associated with lupus.”

Reviewer 2

Comment 1: Overall, this is an interesting manuscript but will require major editing before it is suitable for publication. The genetic association of rare variants in the smaller study by Lee et al did not replicate in the larger study is the strength of this paper. Most of the variants in Lee et al did not have any functional validation, except for the 2 fs mutations which were found to affect localization, replicated in this work. I think that in the current age of large genomic analysis this paper should conclude that functional validation of novel rare variant associations is necessary due to the issues of population stratification. I think the meta-analysis is also strong.

We thank the reviewer for these encouraging comments. We have added the comment on functional validation to the discussion: “Similarly, our functional characterization of these variants suggests few (2/10) of them impair the exonuclease function of TREX1, highlighting the important role of functional validation for candidate lupus risk variants.”

Comment 2: The IFN scores and the MRI associations have many methodological drawbacks that either need to be very clearly stated and included in the limitations or removed.

We thank the reviewer for these comments. We have comprehensively addressed the concerns about IFN scores and describe this in detail in our response to Comment 9 below (and also reviewer 3 comment 5). We have added details about the brain MRI features in the following sections:

Results: “No association was found (Figure 4C; see Supplementary Figure 4 for all individual traits tested), and imaging-derived brain volumes associated with SLE did not differ by carrier status (Figure 4D).”

Methods: “Expanded clinical and radiological phenotypes were defined from autoimmune diseases, common neurological comorbidities of lupus (due to the previous association of TREX1 variants with neurological manifestations in lupus), and radiological phenotypes observed in lupus (white matter hyperintensities and brain atrophy).”

Discussion (limitations section): “Finally, neurological manifestations of SLE are heterogeneous and not limited to the six clinical phenotypes examined here. Brain imaging changes are also heterogeneous in SLE and in our neuroradiological analyses we have analysed both brain volumes as well as white matter hyperintensity volume. The latter is the major structural imaging abnormality in people with SLE, and likely reflects accelerated microvascular disease, but is not seen in all patients.”

We have finally added a description of the demographics in people with versus without SLE and broke down this description by proteomics and MRI availability in an expanded Supplementary Table 3.

Comment 3: The genetic ancestry of the UK Biobank is largely European and may add another degree of important bias that should be addressed in limitations.

We thank the reviewer for this comment and agree that the European ancestry of UK Biobank is a potential limitation. We have added this potential source of bias to the limitations. We have also broken down the analysis for lupus and Sjogren as well as MIRO score by ancestry group (European, South Asian, West African, East Asian) and present this in a new Supplementary Table 12. Although the numbers are low in the three last groups, we observed a higher proportion of lupus/Sjogren cases and *TREX1* carriers in these other ancestries as compared to Europeans, but no *TREX1* carriers were diagnosed with lupus or Sjogren.

We would highlight that in our meta-analysis we have included a large multiancestral cohort, and these findings are consistent.

Results: “Likewise, no ancestry-specific trends were observed for SLE/SjD across individuals of South Asian (n=6,429), West African (n=3,854) or East Asian (n=1,787) ancestries, whether cases were defined from any sources or hospital records only, although samples sizes were lower (Supplementary Table 12)”.

Discussion (last paragraph on limitations): “First, we cannot exclude a predisposition to lupus by some ultra rare *TREX1* variants due to insufficient statistical power, especially in non-European ancestry populations which are underrepresented in UKB. The meta-analysis, however, includes a large study with SLE patients of European American (3,936 cases), Asian (1,265 cases), African American (1,527 cases), Gullah (152 cases) and Hispanic (1,492 cases) ancestries reporting neutral associations within each group, suggesting that any strong ancestry-specific *TREX1*-lupus association is unlikely.”

Comment 4: “We first show that *TREX1* variants 21 do not confer risk for lupus in UK Biobank and that most reported risk variants are functionally neutral 22 in mutagenesis experiments.” Moreover, UKB participants were aged 40-69 years at 62 enrolment in 2006-2010, meaning that the large majority of participants predisposed to SLE will by 63 now have been diagnosed as SLE incidence peaks in middle age.” This is true, but SLE can be diagnosed after enrolment, and this won't be detected in the database. I think the cross-sectional nature of the study is a major limitation that must be mentioned.

We thank the reviewer for this comment and we have made it clearer in the text that, while the cross-sectional nature of the study is a limitation, diagnoses which were both prevalent and incident are captured.

Introduction: “Moreover, UKB participants were aged 40-69 years at enrolment in 2006-2010 and the study captures both incident and prevalent diagnoses, meaning that the large majority of participants predisposed to SLE will by now have been diagnosed, as SLE incidence peaks in middle age.”

Comment 5: the genetic association of rare variants in the smaller study by Lee et al did not replicate in the larger study is the strength of this paper. Most of the variants in Lee et al did not have any functional validation, except for the 2 fs mutations which were found to affect localization, replicated in this work. I think that in the current age of large genomic analysis this

paper should conclude that functional validation of novel rare variant associations are necessary due to the issues of population of stratification?

We have added this comment to the discussion: “Similarly, our functional characterization of these variants suggests few of them (2/10) impair the exonuclease function of TREX1, highlighting the important role of functional validation for candidate lupus risk variants.”

Comment 6: “~469k participants and reliable capture of SLE diagnoses 61 through routinely collected healthcare data .~54k UKB participants, also offers an opportunity to develop and detect 66 oligoprotein interferon signatures which form an important part of the immune endophenotypes 67 associated with SLE.” What is the demographic constitution of the 54k which have proteomic data. Is there a difference in the demographics of this cohort compared to those on whom we have exome data? It is only 10% of the sample. There is a large potential for bias here.

We thank the reviewer for highlighting the potential for selection bias in substudies. We have expanded Supplementary Table 3 to include demographics of participants with vs without proteomic score.

Most participants (87%) in the proteomics project of UKB were selected randomly from baseline participants and were representative of the UKB project¹. A small fraction of participants with proteomics (11%) was chosen to enrich certain conditions such as SLE by the consortium of pharmaceutical companies. At most, this selection has increased the statistical power (optimizing the proportion of participants with lupus who also have proteomics) and is unlikely to have been conditioned on both SLE status and *TREX1* variants (necessary to induce selection bias, whereby people with SLE and *TREX1* variants are enriched and contribute to drive the signal). The absence of signal for SLE risk in *TREX1* variant carriers also argues against this scenario.

Comment 7: “10 coding TREX1 risk variants initially reported⁷ are not associated with SLE⁷¹ or SjD in UKB.” I am curious to why the authors only selected the variants in this study, which is an old association study without much functional validation.

We thank the reviewer for this question. We chose these variants because this paper is the most highly cited example of a link between TREX1 mutations and SLE (Lee Kirsch et al. ~800 citations), and has been influential in the perception of a link between *TREX1* mutations and strong increased risk for SLE (relative risk ~25). For example “SLE risk variants” in ClinVar cite this reference. This paper is often cited as the primary source of evidence of a link between TREX1 mutations and SLE (e.g. Belot et al: PMID: 38263665). Later in the paper we test a broader set of variants.

Comment 8: Page 5, line 92- Please do use the term white for race or European for ancestry and do not use Caucasian: Popejoy AB. Too many scientists still say Caucasian. Nature. 2021;596(7873):463-.

Thank you for highlighting this, together with this article. We have corrected the terminology throughout and used 'European' ancestry when referring to genetic background.

Comment 9: "We therefore sought to develop and validate a proteomic signature 129 of type I interferon activation. We defined 12 Markers of Interferon- α Response in Olink (MIROs) 130 from analytes on the Olink Explore 3072 platform and built a composite oligoprotein measurement (the 131 MIRO score) which averages their standardized levels (Figure 3A)." Did the authors do a training and test set for the MIRO validation? Using a new, not validated measure of Type I IFNs without more detailed validation gives me pause. The proteins selected are IFNs but very few are part of the validated scores most commonly used for SLE. I do not see enough validation to include this in the paper.

Kim H, de Jesus AA, Brooks SR, Liu Y, Huang Y, VanTries R, Montealegre Sanchez GA, Rotman Y, Gadina M, Goldbach-Mansky R. Development of a validated interferon score using NanoString technology. *Journal of Interferon & Cytokine Research*. 2018 Apr 1;38(4):171-85.

Rice GI, Melki I, Frémond ML, Briggs TA, Rodero MP, Kitabayashi N, Oojageer A, Bader-Meunier B, Belot A, Bodemer C, Quartier P. Assessment of type I interferon signaling in pediatric inflammatory disease. *Journal of clinical immunology*. 2017 Feb;37:123-32.

We thank the reviewer for these comments regarding the interferon scores derived from Olink proteomic datasets. We are also grateful for similar comments from Reviewer 3 (comment 5). We agree that it is important to develop and validate this oligoprotein interferon score further, in accordance with the reviewers' suggestions. We have done this in the revised manuscript, including further experimental validation.

- The revised score is developed using proteomic datasets in UK Biobank participants who are being treated with recombinant type I interferon therapies (IFN- α and IFN- β) at the time of blood sampling. This leverages an important natural experiment within UK Biobank.
- Specifically, we have generated the score from proteomic datasets from people being treated with recombinant IFN- β , and internally validated the score against individuals receiving recombinant IFN- α (Note: transcriptomic responses induced by IFN- α and IFN- β are mediated through IFNAR and are very similar PMCID: PMC5159705).
- We have used penalized logistic regression model trained to achieve the best cross-sectional prediction of recombinant type I interferon therapy.
- In addition to internal validation of the score against individuals treated with recombinant IFN- α in UK Biobank, we have performed multiple additional validations, including external validation described below.

Our revised oligoprotein interferon signature comprises three proteins encoded by well-recognised ISGs: SIGLEC1, RIG-I and IFIT3. For example SIGLEC1 is a core component of interferon transcriptomic signatures/scores used in both the papers cited by the reviewer above. RIG-I and IFIT3 which are also widely recognized ISGs, and the latter is widely used in transcriptomic signatures.

In addition to this more robust methodology trained on individuals exposed to recombinant type I IFN, we show multiple independent approaches for validation:

Principal funder:

Medical
Research
Council

- 1) The score is highly elevated in individuals in UKB who are receiving both recombinant IFN- α and IFN- β therapies (revised Figure 3B, below)
- 2) MIRO scores have the strongest correlation with proteins involved in the type I interferon response across other analytes available in UKB-PPP (revised Figure 3C, below)
- 3) We have externally validated the new oligoprotein interferon signature against gold-standard single molecule ELISA for IFN- α in an interferonopathic disorder (Sjogren disease). This analysis is based on paired samples where we have generated both Olink proteomic data and Single molecule ELISA data. (New supplementary Figure 3, below)
- 4) The new oligoprotein interferon signature shows specificity for type I IFN (e.g. IFN- α) and does not correlate with type II IFN (IFN γ). (New supplementary Figure 3, below)

We present the new results in a revised Figure 3.

Results:

“An oligoprotein interferon signature identifies interferonopathic autoimmune diseases

Lupus is considered the prototypical example of a sporadic autoimmune disease associated with prominent activation of the type I interferon response². Early studies linking *TREX1* with SLE also included conditions such as SjD which is similarly associated with elevated type I interferon. This is biologically plausible since *TREX1* is a negative regulator of the type I interferon response. We therefore asked whether we could use UKB-PPP to identify “interferonopathic” autoimmune disorders and test their association with *TREX1* mutations. To date, however, there has been no systematic identification of autoimmune disorders associated with activation of the type I interferon response. This is because quantification of type I interferon in serum has been challenging due to the low abundance of these proteins. Measuring interferon- α is nontrivial and requires high sample volumes, and for this reason is challenging to measure at scale in large cohort studies such as UKB. An alternative surrogate for interferon- α protein concentrations are RNA-based interferon signatures, which are often used as a biomarker in clinical studies of interferonopathic diseases³. However, transcriptomic data are not currently available for UKB. To overcome these challenges, we therefore sought to develop and validate an oligoprotein interferon signature of type I interferon activation derived from broad-capture proteomic datasets. We included 22 correlates of type I interferon levels available in UKB-PPP (Olink Explore 3072 proteomics platform) in a penalized logistic regression model trained to achieve the best cross-sectional prediction of recombinant type I interferon therapy (interferon- β in people with multiple sclerosis at the time of baseline blood sampling, **Figure 3A**). The oligoprotein interferon signature, which we termed the Markers of type I interferon Response in Olink (MIRO) score, was then derived as the sum of the three proteins retained in the model (SIGLEC1, RIG-I and IFIT3) weighted by their penalized coefficient followed by standardization. We tested the score in three validation experiments. Firstly, the oligoprotein interferon signatures are strongly increased in individuals treated with recombinant interferon- α at the time of blood sampling (**Figure 3B**). Secondly, MIRO scores have the strongest correlation with proteins involved in the type I interferon response across other analytes available in UKB-PPP (**Figure 3C**). Thirdly, we

Principal funder:

Medical
Research
Council

externally validated the score in a cohort of individuals with SjD from the United Kingdom Primary Sjogren's Syndrome Registry (UKPSSR) where paired interferon- α single molecule ELISA and Olink proteomics were measured. We found significant correlation between the oligoprotein interferon signature and interferon- α concentrations, but not interferon- γ (Supplementary Figure 3).”

Figure 3. Derivation and validation of an oligoprotein type I interferon signature from Olink proteomics (Markers of type I Interferon Response in Olink or MIRO score) and identification of interferonopathic autoimmune diseases in UK Biobank.

Panel A: Derivation of the MIRO score using Olink Explore 3072 analytes in UK Biobank. Circle packing of included interferon-stimulated genes (SIGLEC1, RIG-I, IFIT3) show area proportional to their absolute coefficient.

Panel B: Association of MIRO with controls and 21 common autoimmune diseases. Cross-sectional design of prevalent conditions (up to 10 years before baseline) in UK Biobank. Positive controls: interferon therapy, chronic viral hepatitis, organ/tissue transplant, HIV infection, liver and kidney disease. Negative control: haemorrhoids. Ten autoimmune conditions had a high (red) or moderate (blue) type I interferon signature (as per point estimates). Statistical significance defined as $p_{adj} < 0.05$ (adjusted for the number of tests within controls and autoimmunity separately).

Panel C: Gene set enrichment analysis of MIRO across biological processes from MSigDB (hallmark) and classical ISGs. Rankings defined using Pearson's correlation coefficients of MIRO with 2,906 Olink analytes and classical ISGs defined using Chaussabel module 3.1.

Partly created with BioRender.com. Abbreviations: CI, confidence interval; HIV, human immunodeficiency virus; ISG, interferon-stimulated gene; LASSO, Least Absolute Shrinkage and Selection Operator; MIRO, Markers of type I Interferon Response in Olink; MSigDB, Human Molecular Signatures Database; p_{adj} , false discovery rate adjusted p-value (Benjamini-Hochberg procedure); SD, standard deviation.

Principal funder:

A

Transcriptomic correlates on INTERFEROME (n=21)										Proteomic correlates in Sjogren (n=5)
AXL	CCL13	CCL8	CD274	CD69	CXCL9	CXCL10	TYMP			
CXCL11	ENPP2	GBP1	IDO1	IFIT1	IFIT3	RIG-I				
IL1RN	IL6	NEXN	SAMD9L	SIGLEC1		LAMP3				
						TNFSF10				

B

Supplementary Figure 3. Derivation and validation of the Markers of type I Interferon Response in Olink (MIRO) score.

Panel A: Candidate correlates of type I interferons included in the penalized logistic regression.
 Panel B: External validation of MIRO in 39 Sjogren disease patients from the UKPSSR. The partial MIRO score was calculated using RIG-I and CD163 (surrogate for SIGLEC1 with correlation between the two proteins in Sjogren disease = 0.673, $p < 0.0001$). SIGLEC1 and IFIT3 were not captured in UKPSSR.

Abbreviations: IFN, interferon; MIRO, Markers of type I Interferon Response in Olink; NPX, normalized protein expression; UKPSSR, United Kingdom Primary Sjogren's Syndrome Registry.

We describe the revised methodology in detail below:

Methods

“United Kingdom Primary Sjogren's Syndrome Registry

The UKPSSR is a national observational cohort of SjD patients fulfilling the 2002 American European Consensus Group classification criteria⁴ and recruited across 30 UK centres with serum samples stored at recruitment (Eppendorf tubes at -80°C)⁵. A random sample of 39 patients from UKPSSR was selected for a serum proteomic substudy quantifying 454 unique analytes with five Olink Target 96 panels (inflammation, immune response, organ damage, cardiovascular Ill and metabolism) and multi-subtype interferon- α levels with a high-sensitivity Simoa assay on an HD-X Analyzer as per manufacturer's instructions (Quanterix, USA). We log-transformed interferon- α levels to obtain a normal distribution for analyses. Participants of the UKPSSR gave written consent for collection of clinical and peripheral blood samples and the research ethics approval was granted by the UK National Research Ethics Committee North West – Haydock.

Derivation and validation of a proteomic type I interferon signature

We sought to develop an oligoprotein interferon score derived from Olink proteomics in the UKB-PPP, which we refer to as Markers of type I interferon Response in Olink (MIRO) score. Because interferon- α and - β are not measured on Olink Explore 3072, we developed this score to capture their downstream proteomic response, analogous to the transcriptomic ‘interferon score’ previously described for monogenic type I interferonopathies which is defined as the median fold change in six canonical ISGs^{6,7}. We first identified gene products quantified on Olink Explore 3072 that correlate with type I interferon levels from either i) a total RNA expression fold change >20x after interferon- α or - β stimulation from a publicly available expression database on ISGs (INTERFEROME v2)⁸ using *in vitro* experiments on human peripheral blood cells under normal conditions, or ii) an ISG (more broadly defined using Chaussabel modules 1.2, 3.4 and 5.12⁹) with a positive association with log-interferon- α at p-value <0.1 in a sample of 39 individuals with SjD from UKPSSR.

We next included these ISGs in a penalized logistic regression model using L1 regularization (Least Absolute Shrinkage and Selection Operator or LASSO regression), which shrinks coefficients to select features and account for collinearity, to achieve the best and most interpretable prediction of type I interferon stimulation. The model was trained on UKB participants with multiple sclerosis treated vs not treated by exogenous interferon- β therapy (any formulation). This class of injectable disease-modifying therapies is used in relapsing-remitting multiple sclerosis and induces a strong transcriptomic response of classical type I ISGs that persists for several days even in long-term users¹⁰. The underlying disease, multiple sclerosis, is not associated with elevated type I interferon, and so the interferon signatures in these patients are driven by the recombinant interferon protein. We considered multiple sclerosis diagnoses either prevalent at baseline or captured within five years of follow-up to balance statistical power and biological relevance as multiple sclerosis pathogenesis precedes diagnosis by 5-10 years¹¹ and because event dates reaching UKB through administrative healthcare databases may lag behind actual diagnostic dates¹². Of 235 participants with multiple sclerosis who had readings on all candidate ISGs and did not have comorbid autoimmunity, chronic kidney disease or liver disease at baseline (to limit potential noise due to other sources of inflammation), we included 21 participants on interferon- β therapy who were randomly matched (1:4 ratio) on age (5-year

Principal funder:

Medical
Research
Council

strata) with 84 participants who did not report using interferon- β at baseline. This sample was used to train and select the most regularized model within one standard error of the regularization parameter minimizing the mean 10-fold cross-validated binomial deviance (λ_{1se}). The MIRO score was calculated as the sum of standardized protein levels for the three included ISGs, each weighted by their L1 penalized coefficient, followed by standardization (mean=0, SD=1; see **Supplementary Table 16** for weights). A score was available for 36,966 UKB participants (mean age at baseline=56.8 years [SD=8.2]; 54.1% females).

We validated the MIRO score in three experiments:

First, we explored the cross-sectional associations at baseline of MIRO with positive control exposures (exogenous interferon- α and - β) and diseases (chronic viral hepatitis¹³, organ/tissue transplant¹⁴, human immunodeficiency virus [HIV] infection¹⁵, liver disease¹⁶ and chronic kidney disease^{17, 18}) and one negative control (haemorrhoids).

Second, we performed a gene set enrichment analysis (GSEA)¹⁹ using Pearson's correlation coefficients of MIRO with 2,906 Olink analytes (with unique UniProt entries) as ranking metric²⁰. We interrogated the 50 hallmark (H) gene sets from the Human Molecular Signatures Database (MSigDB) plus a set of classical ISGs (Chaussabel module 3.1⁹) without restricting gene set sizes (range of genes per set: 5-107). We report normalized enrichment scores (to account for gene set size in enrichment estimates) along with false discovery rate (FDR)-adjusted p-values (p_{adj}) from the Benjamini-Hochberg procedure.

Third, we used a sample of 39 SjD patients with proteomics from UKPSSR to calculate Pearson's correlation coefficients for RIG-I (from Olink) and a partial MIRO score with log-interferon- α (from high-sensitivity Simoa assay) and interferon- γ (from Olink; excluding $n=19$ with levels below the limit of detection). Because SIGLEC1 and IFIT3 were not captured in the UKPSSR Olink Target 96 panels, the partial MIRO score was calculated using CD163 as a surrogate for SIGLEC1 (correlation between the two proteins in SjD: $r=0.673$, $p<0.0001$).

Having validated the score, we next surveyed 21 common autoimmune diseases²¹ to identify type I interferon-related autoimmune conditions using MIRO. In these two last cross-sectional experiments, medical conditions were defined from diagnoses up to 10 years prior to baseline after excluding people with comorbid autoimmunity, prior organ/tissue transplant (except when tested as trait of interest) or on interferon therapy. Associations were tested using linear regressions of MIRO (dependent variable) with each condition of interest adjusted for age (continuous) and sex. Models of medical conditions were also adjusted for interferon- γ levels (to account for its potential association with the type I interferon proteomic signature and traits) and, except for instances with low sample sizes (interferon therapy, dermatomyositis, Addison's disease and myasthenia gravis), adjusted for ethnicity (white vs non-white), body mass index (BMI; continuous), smoking (current, previous, never), alcohol use (four intake levels: none, low, moderate, high²²) and Townsend deprivation index²³ (continuous) at baseline."

Comment 10: Page 18, lines 360-362: "These brain 360 volumes were available for a range of 188-193 participants with SLE/SjD, 42,337-43,621 participants 361 without SLE/SjD, 85-87 TREX1 variant carriers and 42,440-43,727 non-carriers." Again, this is a small subset of the entire cohort. Was an analysis done to determine if they differ from the larger cohorts in important ways which may bias the results?

We provide this analysis in Supplementary Table 3 which shows that the population with vs without data are broadly comparable.

Comment 11: Page 19 lines 388-392- the exclusion criteria seem to be too liberal. Why did they choose a CADD <10, when 12-20 is the current cutoff, and with a population based approach, searching for rare, deleterious variants seems most pragmatic?

Thank you for this comment. We used the CADD score to filter out some potentially mislabeled “pathogenic/likely pathogenic” variants from data sources (ClinVar, LOVD and reviews). This cut-off was chosen to avoid removing known causal variants such as c.150_151del (p.Gln51Glyfs) (CADD=17.6). To avoid excluding deleterious variants, we only removed those fulfilling both a CADD <10 and a MAF \geq 0.1% in UKB, gnomAD or 100GP, which resulted in the exclusion of 2 out of 86 variants (c.679G>A and c.720G>C), both of which were already included and assessed in the first analysis of “previously reported TREX1 lupus risk variants” with neutral results (no lupus cases for both variants) (see Figure 1).

Most included variants had CADD >20: of the 84 TREX1 variants included from our systematic search, only 3 (3.6%) had a CADD <10, and 11 (13.1%) had a CADD <20. When we tested the impact of predicted deleteriousness (by LOFTEE and CADD), we did not observe an effect of these annotations on the association with interferon signatures (Figure 5). This suggests that the CADD threshold had no meaningful impact on our results.

We additionally provide a more detailed Supplementary Table 9, showing the CADD score and the number of participants with lupus and MIRO score (along with sample sizes) for each variant.

Comment 12: The section of the paper examining TREX 1 variants with interferonopathic autoimmunity seems to be a different story. If the paper wants to look at all causes of Type I IFNopathies, they should assess it, but including these here seems to dilute the paper. Again, the choice of variants to assess may not be stringent enough and bias the results to the see no association.

We thank the reviewer for this comment regarding the narrative flow of the paper, in particular our lack of clarity in explaining how the proteomic aspect of the UKBiobank study offers additional strengths in unpicking the relationship between TREX1, type I interferon and SLE-related phenotypes. This is particularly important because the role of TREX1 as a negative regulator of type I interferon underlies its original choice as an SLE candidate gene.

In the revised manuscript we have explained more clearly the rationale for our approach. There are two reasons we integrated oligoprotein interferon signature-associated phenotypes into this study.

Firstly, we wished to expand our UKB genetic association study beyond SLE to other autoimmune disease with prominent interferon activation, because this is the approach used in the influential Lee-Kirsch *Nature Genetics* 2007 paper. Specifically, a Sjogren syndrome cohort was included in that paper because of its phenotypic overlap with SLE, including elevated type I interferon. A *TREX1* mutation was reported in that Sjogren disease cohort which strengthened the disease

association. Our approach also includes SjD, but also includes a wider analysis to include other related interferon-associated autoimmune phenotypes, which we systematically identify using interferon signatures. This allows us to address the issue of phenotypic overlap (raised in the original Lee-Kirsch paper) in a more systemic and comprehensive way.

Secondly, the immune dysregulation that characterises SLE is detectable in serum years prior to diagnosis (Arbuckle NEJM 2003), and can identify individuals at high risk of SLE. As such interferon signatures are likely to represent an important intermediate phenotype (and in some individuals represent a “forme fruste” of SLE). In the revised manuscript we have demonstrated this and provide clear evidence that the presence of oligoprotein signatures in UKB identify individuals at future risk of a clinical diagnosis of SLE. We have done this by performing a time to event analysis of interferon signatures measured at baseline, showing increased risk of incident diagnosis of SLE. As such, our analysis that not only is risk of SLE diagnosis not elevated in *TREX1* mutation carriers, but interferon signatures (which may herald SLE diagnosis in the decade ahead) are not present either.

We have expanded the results section to better justify our exploration of related clinical and immune phenotypes:

Results 1: “Early studies linking *TREX1* with SLE also included conditions such as SjD which exhibits phenotypic overlap with SLE and is similarly associated with elevated type I interferon. This is biologically plausible since *TREX1* is a negative regulator of the type I interferon response. We therefore asked whether we could use UKB-PPP to identify “interferonopathic” autoimmune disorders and test their association with *TREX1* mutations.”

Results 2: “Serological evidence of SLE can precede diagnosis by many years. Indeed, in UKB-PPP, we found that elevated type I interferon signatures at baseline in people without an SLE diagnosis were strongly associated with future risk of SLE diagnosis (**Figure 5A**; see **Supplementary Table 9** for all estimates). Interferon signatures could be reliably detected up to 9 years prior to diagnosis (**Supplementary Figure 5**). We therefore explored whether *TREX1* variant carriers might have evidence of peripheral type I interferon activation. We found no difference in MIRO scores between carriers and non-carriers of *TREX1* variants (**Figure 5B**; see **Supplementary Figure 6** for individual component analytes).”

Figure 5. Oligoprotein interferon signatures are associated with high risk of SLE in UKB-PPP, but are not elevated in TREX1 mutation carriers

Panel A: A peripheral proteomic signature for type I interferon activation (MIRO score) predicts incident SLE (left) but not multiple sclerosis (right). Adjusted time-to-event curves by quartile of MIRO score from Cox regression models (HRs with 95% CIs).

Panel B: Density plot of MIRO score by TREX1 genotype.

Panel C: Density plot of MIRO score by LOFTEE classification.

Panel D: Scatter plot of MIRO score by CADD and LOFTEE classification.

Abbreviations: CADD, Combined Annotation-Dependent Depletion score (PHRED-like scaled); CI, confidence interval; HR, hazard ratio; pLOF, high-confidence predicted loss-of-function from LOFTEE; MIRO, Markers of type I Interferon Response in Olink.

Supplementary Figure 5. Detection of oligoprotein interferon signatures before diagnosis of lupus in UK Biobank.

Panel A: Distribution of MIRO scores relative to recorded SLE diagnosis in UK Biobank. The blue dotted line at $y=0$ indicates the mean MIRO score in UKB. Smooth curve produced by locally estimated scatterplot smoothing (LOESS) along with standard errors (grey).

Panel B: Analysis of MIRO scores by 3-year epoch prior to SLE diagnosis in UK Biobank. Participants with SLE were matched 1:10 to controls selected at baseline on age and sex. Plot shows mean MIRO scores with 95% confidence intervals by group and p-values obtained through linear regressions. Intervals are closed to the left and open to the right, i.e. -15: [-15, -12), -12: [-12, -9), -9: [-9, -6), -6: [-6, -3), -3: [-3, 0).

Abbreviations: MIRO, Marker of type I Interferon Response in Olink; SLE, systemic lupus erythematosus.

Updated methods: “We used Cox proportional hazards models to test the association of MIRO scores at baseline with incident diagnoses (SLE and multiple sclerosis) adjusted for potential predictors of autoimmunity (age, sex, ethnicity, smoking, Townsend deprivation index, presence of other autoimmune diseases). For each condition, participants without prevalent diagnosis were included at baseline and followed until the first of outcome occurrence, death, loss of follow-up or study end date (31/OCT/2022; last day of the month with mostly complete data for National Health Service [NHS] England inpatient records, the main source of diagnoses).”.

Comment 13: Page 22, line 458: We replicated these neutral associations with RF and CRP, two blood biomarkers associated 458 with lupus. These are not common lupus biomarkers. ANA and ESR are much more commonly associated with lupus. Overall, I think adding someone with rheumatology expertise and specifically lupus expertise to the study team would greatly enhance the manuscript.

We thank the reviewer for this relevant comment as well as review 3 (comment 11) for raising a similar question.

We agree that CRP and RF are not lupus biomarkers commonly used in clinic and that ANA and ESR would have been preferable. The latter, however, were not available in UKB. In light of these comments and those of reviewer 3, we agree that this section could be misinterpreted, and we have removed it.

We would reassure that reviewer that the lead author, although a neurologist, has substantial lupus expertise, leading joint clinics and research in the area for over a decade. We would highlight the inclusion of two collaborators to the revised manuscript, including a rheumatologist with expertise in autoimmunity (Professor Fai Ng, University of Cork) and a lupus nephrologist Professor Neeraj Dhan (University of Edinburgh). These authors have been involved in the validation of the MIRO score, but have also contributed to redrafting of the manuscript.

Comment 14: The GitHub link provided did not work for me to review the code.

Please find the link here: https://github.com/barioux/UKB_TREX1/tree/main

This link was also added in the “Code availability” section.

Reviewer 3

Comment 1: The paper lacks a section on its limitations, which should be added, along with addressing other matters listed below.

We thank the reviewer for this comment. We have added this as a dedicated section, and cover amongst other issues: very rare variants, other ancestries, age at SLE diagnosis and diagnostic accuracy.

Comment 2: Medical record diagnosis is notoriously weak, in particular in relation to heterogeneous autoimmune diseases where both under- and over-diagnosis are common. 35% in total of cases had the diagnosis based on a primary care-assigned diagnosis, which in this reviewer's experience is likely to represent a significant proportion of false positive SLE diagnoses, weakening any associations.

We thank the reviewer for this comment and address this in the revised manuscript.

Among UKB participants with lupus, 241 (18.5%) were diagnosed from primary care records without diagnostic codes from hospital records. We should note that these diagnoses may be either from a GP or from outpatient secondary care as diagnoses following referrals are generally reported back and recorded by community practices. In UKB, most lupus cases (69.5%) originated from inpatient hospital records with high positive predictive value (>95%). In addition, the meta-analysis included a well phenotyped cohort of >6k cases fulfilling the American College of Rheumatology criteria.

We have expanded the description of the data sources for primary care records in the Methods: "Primary care data include both diagnoses from general practitioners and those from outpatient secondary care (e.g., rheumatology clinics) as these are generally reported back to the practice."

We have also added the potential for misclassification as a limitation in the discussion (limitations section): "Third, lupus diagnoses from administrative health records are prone to misclassification. The majority of UKB cases, however, were found through inpatient hospital records which have a high positive predictive value (>95%), and the neutral meta-analysis included a well phenotyped cohort fulfilling the American College of Rheumatology criteria."

We finally re-analyzed lupus and Sjogren disease by defining cases only from hospital records, which yielded similar results (presented in the new Supplementary Table 12).

Comment 3: Ancestry linkage studies are informative, suggesting a possible reason for previous reported associations between TREX variants and SLE. However, UKB is hugely dominated by white/European ancestry subjects, known to have the lowest prevalence and severity of SLE. Significant missing ancestries include East Asian ancestries, in which with a very high prevalence of young onset SLE may have a different result. African ancestries are also <>10% of the cohort.

Principal funder:

Medical
Research
Council

Thank you for this comment. We have answered this point in reviewer 2 comment 3 above. Briefly, we have highlighted this in the limitations section and described how prior studies with multi-ancestral structures included in the meta-analysis do not support an association.

Comment 4: The cohort is also light-on for young onset disease, where genetic associations are believed to be stronger. <2% of cases were diagnosed at age <20.

This is an important point which is addressed in the response to reviewer 1 comment 3 above. This is now addressed in the limitations section.

Comment 5: The so-called MIRO protein signature is entirely derived from public domain sources, namely mRNA expression data after IFN-alpha stimulation, and predicted secretion of the gene products. This is a far from ideal way to develop such a signature, which ideally would be based on detection of proteins from human cells exposed to IFNAR activation, which in turn would not be limited to IFN-alpha stimulation as multiple other type I IFNs are implicated in SLE. Earlier work such as that of Bauer et al (10.1371/journal.pmed.0030491.st002) combined measurement of proteins directly in patient samples with in vitro exposure of cells to IFN in order to increase confidence that the signature was valid; that has not been done here. Experimental replication of this signature is required, and its lack is a significant caveat.

We thank the reviewer for this important comment. As outlined in our response to reviewer 2 comment 9, we have developed our methodology to address these concerns.

We also agree with the reviewer that our approach has limitations, and we address this explicitly in the discussion section of the revised manuscript.

Discussion: “The UKB-PPP offers an important opportunity to study the link between type I interferon, SLE and *TREX1* mutations. Interferon- α is typically not captured in proteomic datasets, and we show here that an oligoprotein signature can be derived from broad capture proteomics which can act as a reasonable proxy measure of interferon- α . This allows us to leverage the additional strength of the UKB-PPP study to evaluate whether *TREX1* mutations confer risk for other “interferonopathic” autoimmune diseases, as well as look for the presence of interferon signatures in mutation carriers. We show that interferon signatures can precede SLE diagnosis for many years, and are observed in a number of other autoimmune conditions, but such type I interferon endophenotypes are not associated with *TREX1* mutations. Given the utility of transcriptomic interferon signatures in the study of lupus, we note the potential for future refinement of this oligoprotein approach for the study of interferonopathic disorders in large deeply phenotyped cohorts such as UKB.”

Comment 6: Similarly, the ‘shoulders’ of individual MIRO protein levels greater than those in the health population are modest (Suppl Figure 4), notwithstanding the small p values for the difference in means potentially driven by the large sample size, though somewhat more impressive for the composite MIRO score.

We thank the reviewer for this comment and highlight that the development of composite scores for detection of interferon signatures has historically provided a more robust outcome measure than single ISGs – and we replicate this with the proteomic signature. Our revised methodology also further

strengthens the “shoulders” (see Figure 5 and Supplementary Figure 6). We would also highlight that the plots showing NPX values (the units used by Olink proteomics) are on the log₂ scale which may attenuate differences (as compared to a non-transformed scale).

Comment 7: The meaning of Figures 3 C&D is unclear.

Panels in Figure 3C showed how autoimmune diseases can be divided into “interferonopathic” and “noninterferonopathic” disorders. This is displayed as packed circles, area proportional to the number of individuals in UK Biobank. We have revised this approach to visual display of information since we agree many readers may not be familiar with this approach.

In the revised Figure 3, we have dropped this circle packing and have kept an improved forest plot showing diseases with a high, moderate and lower type I interferon signature (Figure 3B).

Comment 8: Figure 4C appears to be monochromatic but the legend refers to red and black components.

In the legend we are referring back to Figure 3B but in the revised version we have made this clearer and apologise for the lack of clarity.

Figure 4C legend: “Autoimmune diseases grouped by type I interferon signature from MIRO scores (Figure 3B)”.

Comment 9: It is not immediately clear what data are represented in Figure 5B – I take it the dots, box, and whiskers are individual MIRO proteins? Each is counted when >2SD - above what?

We thank the reviewer for this comment. These plots were showing the proportion of participants with protein values > 2SD of the mean for each Olink analyte included in the MIRO score, showing that participants with lupus had more often extreme values across these analytes than people without lupus but that this was not the case for *TREX1* carriers. We have refined the MIRO score and now with only 3 components these plots are not as informative and we decided to remove them as they duplicate information from the revised Supplementary Figure 5.

Comment 10: Noting that IFNs are generally low abundance in human serum samples, they are detectable in some individuals. Were any type I IFN proteins detected in any sample? Did they relate to *TREX* variants?

Thank you for this comment, which gets to the heart of the challenges of measuring IFNa in large scale human studies. The low abundance of IFNa in human blood has always meant accurate measurement can only be performed using single molecule ELISA. Furthermore IFNa is not captured using Olink proteomics. UKB has strict release criteria for stored serum and, given that our Simoa assay uses ~200uL of volume per datapoint, it is unlikely that we are able to access this depletable resource because of the large volumes currently required. Hence we have validated the MIRO score against Simoa to demonstrate that this is a reasonable proxy measure.

We have added an explanation in the Results: “Measuring interferon- α is nontrivial and requires high sample volumes, and for this reason is challenging to measure at scale in large cohort studies such as UKB. An alternative surrogate for interferon- α protein concentrations are RNA-based interferon signatures, which are often used as a biomarker in clinical studies of interferonopathic diseases. However, transcriptomic data are not currently available for UKB. To overcome these challenges, we therefore sought to develop and validate a proteomic signature of type I interferon activation.”

Comment 11: CRP is an unusual biomarker to choose in the setting of SLE, as in most studies it has little or no correlation with disease activity and is commonly measured in the reference range in people with SLE, even when active. I do not recognise this as a valid reference biomarker for the research question being addressed here and recommend deletion of this section.

Please refer to comment 13 of reviewer 2 on this point.

Comment 12: In vitro variant testing was done in murine cells – does this risk missing functionally important effects restricted to human cells?

TREX1 nuclease function and localisation is highly conserved from human to mouse. While it is possible there are downstream effects our testing is restricted to enzymatic location and ssDNA nuclease activity which are conserved across mouse and human. The TREX1-null mouse cell assay is well accepted for these purposes. We have also tested a small subset of variants in human cells and the effects on nuclease function and localisation are the same.

We have expanded on this in the Methods: “Both ssDNA nuclease activity and TREX1 localisation are similar across mouse and human species”.

Comment 13: The authors appear to be from a neurosciences centre, but apart from this why were CNS manifestations of SLE chosen to be called out?

The author is correct that the research group has an interest in neuroimmunological conditions, including neurolupus. While the main focus is on SLE, we did focus CNS manifestations for two reasons: (i) biallelic loss of function in *TREX1* leads to AGS, an interferonopathy with mainly CNS manifestations, and (ii) the findings of Namjou et al showed a potential association between a *TREX1* haplotype and CNS manifestations of lupus. In the experience of our centre, some clinicians perform *TREX1* sequencing in lupus patients with atypical CNS manifestations (see comment 14 below). We have stated this rationale more clearly in the revised manuscript.

Results: “We used this proteomic signature approach to test a broader hypothesis that carriers of any disease-causing *TREX1* variants have an increased risk of type I interferon-related disease or neurological comorbidities of SLE (as suggest by a previous study).”.

Methods: “Expanded clinical and radiological phenotypes were defined from autoimmune diseases, common neurological comorbidities of lupus (due to the previous association of *TREX1* variants with neurological manifestations in lupus), and radiological phenotypes observed in lupus (white matter hyperintensities and brain atrophy).”

Comment 14: The discussion is too long. TREX sequencing is not part of the diagnosis or clinical care of SLE.

Thank you for this comment.

We agree with this comment (and also note that we have increased the “limitations” section of the discussion in view of reviewer comments). We have therefore removed sections of the discussion which were less directly relevant to the main findings of the paper, and hope this leads to a more focused and readable discussion. Deleted text is marked in blue on the revised manuscript.

We address *TREX1* sequencing in the comment above. We have removed from the abstract “~~At a practical level, we suggest there is no merit in sequencing *TREX1* in patients with lupus, where monogenic disease is not suspected.~~”

References

1. Sun BB, Chiou J, Traylor M, et al. Plasma proteomic associations with genetics and health in the UK Biobank. *Nature* 2023;622:329-338.
2. Banchereau J, Pascual V. Type I Interferon in Systemic Lupus Erythematosus and Other Autoimmune Diseases. *Immunity* 2006;25:383-392.
3. Rice GI, Forte GM, Szykiewicz M, et al. Assessment of interferon-related biomarkers in Aicardi-Goutieres syndrome associated with mutations in *TREX1*, *RNASEH2A*, *RNASEH2B*, *RNASEH2C*, *SAMHD1*, and *ADAR*: a case-control study. *Lancet Neurol* 2013;12:1159-1169.
4. Vitali C. Classification criteria for Sjogren's syndrome: a revised version of the European criteria proposed by the American-European Consensus Group. *Annals of the Rheumatic Diseases* 2002;61:554-558.
5. Ng W-F, Bowman SJ, Griffiths B, on behalf of the Usg. United Kingdom Primary Sjögren's Syndrome Registry—a united effort to tackle an orphan rheumatic disease. *Rheumatology* 2011;50:32-39.
6. Rodero MP, Tesser A, Bartok E, et al. Type I interferon-mediated autoinflammation due to DNase II deficiency. *Nature Communications* 2017;8:2176.
7. Burska A, Rodríguez-Carrio J, Biesen R, et al. Type I interferon pathway assays in studies of rheumatic and musculoskeletal diseases: a systematic literature review informing EULAR points to consider. *RMD Open* 2023;9:e002876.
8. Rusinova I, Forster S, Yu S, et al. INTERFEROME v2.0: an updated database of annotated interferon-regulated genes. *Nucleic Acids Research* 2013;41:D1040-D1046.
9. Chaussabel D, Quinn C, Shen J, et al. A Modular Analysis Framework for Blood Genomics Studies: Application to Systemic Lupus Erythematosus. *Immunity* 2008;29:150-164.
10. Feng X, Bao R, Li L, Deisenhammer F, Arnason BGW, Reder AT. Interferon- β corrects massive gene dysregulation in multiple sclerosis: Short-term and long-term effects on immune regulation and neuroprotection. *eBioMedicine* 2019;49:269-283.
11. Tremlett H, Marrie RA. The multiple sclerosis prodrome: Emerging evidence, challenges, and opportunities. *Multiple Sclerosis Journal* 2021;27:6-12.
12. Clifton L, Liu X, Collister JA, Littlejohns TJ, Allen N, Hunter DJ. Assessing the importance of primary care diagnoses in the UK Biobank. *Eur J Epidemiol* 2024;39:219-229.
13. Bolen CR, Robek MD, Brodsky L, et al. The blood transcriptional signature of chronic hepatitis C virus is consistent with an ongoing interferon-mediated antiviral response. *J Interferon Cytokine Res* 2013;33:15-23.
14. Rascio F, Pontrelli P, Accetturo M, et al. A type I interferon signature characterizes chronic antibody-mediated rejection in kidney transplantation. *The Journal of Pathology* 2015;237:72-84.
15. Mackelprang RD, Filali-Mouhim A, Richardson B, et al. Upregulation of IFN-stimulated genes persists beyond the transitory broad immunologic changes of acute HIV-1 infection. *iScience* 2023;26:106454.
16. Hackstein C-P, Spitzer J, Symeonidis K, et al. Interferon-induced IL-10 drives systemic T-cell dysfunction during chronic liver injury. *Journal of Hepatology* 2023;79:150-166.
17. Zoccali C, Vanholder R, Massy ZA, et al. The systemic nature of CKD. *Nature Reviews Nephrology* 2017;13:344-358.
18. Anders H-J, Lichtnekert J, Allam R. Interferon-alpha and -beta in kidney inflammation. *Kidney International* 2010;77:848-854.
19. Subramanian A, Tamayo P, Mootha VK, et al. Gene set enrichment analysis: a knowledge-based approach for interpreting genome-wide expression profiles. *Proc Natl Acad Sci U S A* 2005;102:15545-15550.

Principal funder:

Medical
Research
Council

20. Zyla J, Marczyk M, Weiner J, Polanska J. Ranking metrics in gene set enrichment analysis: do they matter? *BMC Bioinformatics* 2017;18:256.
21. Conrad N, Verbeke G, Molenberghs G, et al. Autoimmune diseases and cardiovascular risk: a population-based study on 19 autoimmune diseases and 12 cardiovascular diseases in 22 million individuals in the UK. *Lancet* 2022;400:733-743.
22. Smyth A, Teo KK, Rangarajan S, et al. Alcohol consumption and cardiovascular disease, cancer, injury, admission to hospital, and mortality: a prospective cohort study. *The Lancet* 2015;386:1945-1954.
23. Nagar SD, Jordan IK, Mariño-Ramírez L. The landscape of health disparities in the UK Biobank. *Database* 2023;2023.

Principal funder:

Medical
Research
Council

Oligoprotein type I interferon signatures, but not *TREX1* variants, increase risk of systemic lupus erythematosus in UK Biobank

Many thanks for the encouraging response to our revised manuscript. We are pleased that the reviewers felt our manuscript was improved. Please find below a point-by-point rebuttal to the constructive comments of the reviewers, and a fully revised manuscript.

We hope the reviewers find that we have addressed their comments.

Colour code

Black = reviewer comments

Purple = author response

Green = new text in revised manuscript

Reviewer 1

Comment 1: The authors' response has reduced some of my concerns, but several points warrant further discussion. The UKB and meta-analyses remain based on gene-level burden analyses, where simple gene burden tests may dilute true signals with numerous neutral variants, despite inclusion of diverse ethnicities (predominantly European).

Moreover, genuine pathogenic variants such as p.Asp130Asn may not have extremely high ORs but could still contribute to sporadic SLE with elevated serum IFN α (DOI: 10.1016/j.clim.2021.108732).

These are missense mutations rather than established dominant-negative mutations like D18N and D200N. The discussion of these critical points seems to be insufficient, with the authors instead vaguely claiming that the meta-analysis has adequate power without addressing the fundamental methodological limitations of burden testing in detecting variant-specific effects.

We thank the reviewer for these comments and are happy to discuss these points further. As requested in the new revision we discuss further the methodological limitations of burden testing in detecting variant-specific effects (see below).

On the point of the mutation raised by the reviewer, we agree that very rare deleterious *TREX1* mutations are occasionally reported in people with SLE, although the disease-relevance of these mutations is not clear. As highlighted by the reviewer, the p.Asp130Asn is a good example. This is a pathogenic/deleterious mutation which can cause Aicardi-Goutieres syndrome when inherited with another loss of function *TREX1* mutation (Yanick Crow, personal communication). UK Biobank is underpowered to examine the SLE risk of individuals with this mutation (there are only 6 individuals with this mutation in UKB, none of whom have a diagnosis of SLE). However in our wider analysis we have tested the hypothesis that AGS *TREX1* mutation carriers are predisposed to SLE (Figure 4), and we found no association. We highlight the need for caution before attributing disease-relevance to such mutations.

Discussion:

“First, we cannot exclude a predisposition to SLE by some *TREX1* variants due to insufficient statistical power (e.g., very rare variants or rare variants associated with disease with small effect sizes). Our UKB study, systematic review and meta-analysis use gene-level analyses and demonstrate a lack of association between *TREX1* variants and SLE. There is the potential for simple gene burden tests to dilute true signals with numerous neutral variants, which can occur despite the inclusion of diverse ancestries in the meta-analysis. It is

Principal funder:

Medical
Research
Council

established that very rare and highly functional deleterious loss of function mutations in *TREX1* can cause monogenic interferonopathic disease through a dominant negative mechanism (e.g., c.52G>A [D18N] and c.598G>A [D200N]), and there are other variants implicated in atypical interferonopathic syndromes with overlap with lupus, such as c.388G>A (D130N) where disease-relevance remains unclear and cannot be addressed in this study because of the rarity (there are only 6 carriers of c.388G>A in UKB, none with a diagnosis of SLE). Furthermore, non-European ancestry populations are underrepresented in UKB. The meta-analysis, however, includes a large study with SLE patients of European American (3,936 cases), Asian (1,265 cases), African American (1,527 cases), Gullah (152 cases) and Hispanic (1,492 cases) ancestries reporting neutral associations for previously reported risk variants within each group¹, suggesting that any strong ancestry-specific *TREX1*-SLE association is unlikely. These analyses highlight the importance of stringent functional assessments to examine causality in future studies of very rare *TREX1* variants that our study has limited power to dissect.”

Reviewer 2

Comment 1: The authors have revised the manuscript, which I still believe has merit and has findings that contribute to the body of literature on the role of *TREX1* variants in SLE. However, I find that there continue to be methodological issues which have not been fully addressed. I think the authors need to clarify that heterozygous *TREX1* variants are not equal, and while some contribute to SLE others do not and finding a rare variant in *TREX1* requires functional evidence before considering causality. This should be reflected in the title.

We are happy to provide this clarification below. We have also commented on this issue in reviewer 1, comment 1, and have modified the discussion to further highlight the role of functional assessments. Our title is already quite long (and it is quite difficult to capture this nuance in a few words), but we would be happy to amend upon editorial request.

Discussion:

“These results have implications for attributing disease relevance of heterozygous *TREX1* mutations found in people with SLE, because our results show that this association is weaker than previously thought. Firstly, *TREX1* variants with no functional consequence are unlikely to be disease relevant. Secondly, monoallelic variants which do not lead to profound functional perturbation (such as seen in dominant negative nuclease-dead mutations like c.52G>A [D18N]) are unlikely to confer increased risk of SLE. This observation may also be relevant for counselling of parents of children with AGS, who have monoallelic loss of function *TREX1* mutations and no demonstrable increased SLE risk. Finally, there remains some very rare deleterious mutations where we do not have enough information to attribute nor exclude a causal role in SLE.”

Comment 2: I do not think the authors have sufficiently addressed the weakness of SLE diagnosis from medical record as a limit.

We thank the reviewer for this comment on diagnostic accuracies in large biobanks.

We acknowledge that medical diagnoses from administrative healthcare records are subject to various degrees of misclassification. Diagnostic accuracy studies have previously reported high positive predictive value (>95%)² for SLE using International Classification of Diseases (ICD) codes from inpatient hospital episodes such as those used to define most cases in UKB. To explore whether misclassification might have led to bias towards the null, we have performed sensitivity analyses by source of diagnosis in Supplementary Table 12, showing no enrichment in lupus diagnosis among *TREX1* variant carriers for inpatient hospital diagnoses (less prone to misclassification). In addition, type I interferon signatures, which are not subject to diagnostic misclassification, were not associated with *TREX1* variants.

We have added a further clarification on the impact of this misclassification in the limitations: “Third, diagnosis of SLE in UKB is based on administrative healthcare records, which can be subject to various degrees of misclassification. This contrasts with more formal genetic studies where more stringent diagnostic criteria are used. SLE diagnoses from administrative health records are prone to misclassification leading to bias towards

Principal funder:

Medical
Research
Council

the null. The majority of UKB cases, however, were found through inpatient hospital records which have a high positive predictive value (>95%), and the neutral meta-analysis included a well phenotyped cohort fulfilling the American College of Rheumatology criteria. Furthermore, type I interferon signatures, which are not subject to diagnostic misclassification, were not associated with *TREX1* variants.

Comment 3: The meta-analysis is based on 2 very old studies with 2 very different approaches to sequencing (one is an Illumina chip assay, which is used for the large, diverse cohort, and may miss many rare novel variants in *TREX1*. This assay would be the weakest to detect variants in non-europeans populations) This needs to be stated as a limitation.

We thank the reviewer for this comment and we now state this as a limitation. We acknowledge that the study by Namjou et al (2011) included in our systematic review does not capture all rare *TREX1* variants and used genotyping array chips to generate data.

The authors examined a focused set of previously reported rare “risk variants”, which we believe are most relevant because of their reported association with lupus. In addition, although performed on genotyping array chips, the authors used stringent variant- and sample-level quality control filters to ensure calls were accurate. This study therefore takes a focused approach to replicate positive associations from Lee-Kirsch et al (2007) and reports neutral results.

We have clarified the scope of variants tested in this study in the systematic review: “The second study (Namjou et al, 2011)¹ aimed to replicate findings from this first study and expand the association of *TREX1* variants with SLE in multi-ancestral cohorts using array genotyping for previously reported rare variants (i.e., not capturing all rare variants) and more common tag single nucleotide polymorphisms (SNPs).”

We have also added a clarification in the limitations to help interpret findings: “The meta-analysis, however, includes a large study with SLE patients of European American (3,936 cases), Asian (1,265 cases), African American (1,527 cases), Gullah (152 cases) and Hispanic (1,492 cases) ancestries reporting neutral associations for previously reported risk variants within each group, suggesting that any strong ancestry-specific *TREX1*-SLE association is unlikely.”

Comment 4: The Miro validation is not sufficient. The use of a cohort of 39 patients as a validation cohort is not sufficient. Also, to use a partial MIRO score which substitutes CD163 for SIGLEC1 is not robust methodology. The authors claim that many of the markers used are similar to validated scores does not substantiate the use of an unvalidated score. Also, the p value and r score for RIG-I are marginal in this validation. They either need to do a large validation of the method or drop this section of the manuscript.

We thank the reviewer for these comments. In our original manuscript we provided extensive internal validation of the MIRO score (see below) and in our last revision we provided new external validation, which demonstrated that the MIRO score is a reasonable, albeit imperfect, proxy for IFN α .

We agree that if the score were to be widely used as a clinical tool, then more extensive external validation would be needed, and we have clarified this in the revision. Transcriptomic interferon scores typically undergo extensive validation in multiple large cohorts if they are to be used in a clinical setting for diagnostic purposes e.g. Rice et al. 2013 Dec;12(12):1159-69. Indeed we will be seeking to develop this methodology in this way, but this is beyond the scope of the current paper, and is not feasible in the timescale for revision (4 weeks).

It is important to emphasise we are not claiming that the score has clinical utility at this stage of its development, rather that the MIRO score is a reasonable proxy for IFN- α and can act as an intermediate (and strongly biologically relevant) phenotype for the purpose of this study. For the stated limited experimental purposes of our paper we feel that the validation of our oligoprotein score is sufficiently robust. We have validated the score in multiple different and complementary ways, both internally and externally.

Our other validation approaches include the following:

1. The score is highly elevated in individuals in UKB who are receiving both recombinant IFN- α and IFN- β therapies (Figure 3B)
2. MIRO scores have the strongest correlation with proteins involved in the type I interferon response across other analytes available in UKB-PPP (Figure 3C)
3. The new oligoprotein interferon signature shows specificity for type I IFN (e.g. IFN- α) and does not correlate with type II IFN (IFN- γ) (Supplementary Figure 3)
4. The score correctly classifies healthy controls, multiple sclerosis as “non-interferonopathic” and SLE and Sjogren disease as “interferonopathic”. These classifications have been confirmed by prior single molecule ELISA studies in cohorts of these patients (summarized in doi: 10.1016/j.immuni.2024.05.017).

As such the MIRO score is sensitive enough to capture elevations in peripheral type I interferons in carriers of *TREX1* variants to test our null hypothesis (H_0 : *TREX1* variants are not associated with an elevation in type I interferons). The MIRO score was externally validated in a sample sufficiently large to capture moderately strong correlation coefficients such as seen for MIRO and IFN- α . This is reflected in confidence intervals that are narrow enough to practically exclude small and modest correlations ($r=0.57$; 95% CI: 0.31, 0.75, $p=0.0001$). Therefore, although we agree that increasing the sample size would reduce statistical uncertainty, we do not believe this would change our interpretation of the data or conclusions.

Revision 1

We have added a substantial new section highlighting the strengths, limitations and future opportunities for oligoprotein interferon signatures:

“The UKB-PPP offers an important opportunity to study the link between type I interferon, SLE and *TREX1* mutations. Interferon- α is typically not captured in proteomic datasets, and we show here that an oligoprotein signature can be derived from broad capture proteomics which can act as a reasonable proxy measure of interferon- α . This allows us to leverage the additional strength of the UKB-PPP study to evaluate whether *TREX1* mutations confer risk for other “interferonopathic” autoimmune diseases, as well as look for the presence of interferon signatures in mutation carriers. We have developed these oligoprotein interferon signatures from multiple sclerosis patients in UKB receiving recombinant type I interferon therapies, which are known to strongly upregulate interferon response genes⁴⁰. This identifies proteins (SIGLEC1, RIG-I, and IFIT3) which are encoded by established ISGs. We have validated the MIRO score both internally and externally against interferon- α , in both patients receiving recombinant interferon- α therapy (in UKB) and in patients with high interferon- α plasma concentrations (in UKPSSR). This shows that the MIRO score is a reasonable yet imperfect proxy for interferon- α concentrations in UKB. We show that oligoprotein interferon signatures can precede SLE diagnosis for many years, and are observed in a number of other autoimmune conditions, but such type I interferon endophenotypes are not associated with *TREX1* mutations. Given the utility of transcriptomic interferon signatures in the study of lupus, we note the potential for future refinement of this oligoprotein approach for the study of interferonopathic disorders in large deeply phenotyped cohorts such as UKB. Further development and refinement of this oligoprotein signature approach, in particular cross-validation with transcriptomic interferon signatures across multiple disease and control cohorts will be an important area for future study, especially if clinical applications of the signature are considered²⁰.”

Revision 2

In our revision we provide new data demonstrating that all three proteins are encoded by strongly upregulated ISGs, which further strengthens the biological underpinning of the score. We provide data from 177 Sjogren Disease patients (where both Simoa and transcriptomic data were available) which shows that all three genes encoding MIRO components are strongly upregulated in UKPSSR, confirming that SIGLEC1, RIG-I and IFIT3 are all ISGs.

Principal funder:

Medical
Research
Council

These new data are presented in the new **Supplementary Figure 3:**

A: Candidate correlates of type I interferons included in the penalized logistic regression.

B: Genes encoding the MIRO score are interferon-stimulated genes. Components of the MIRO score correlate significantly with interferon- α levels in UKPSSR.

C: External validation of MIRO in 39 Sjogren disease patients from UKPSSR. The partial MIRO score was calculated using RIG-I and CD163 (surrogate for SIGLEC1 with correlation between the two proteins in Sjogren disease = 0.673, $p < 0.0001$). SIGLEC1 and IFIT3 were not captured in UKPSSR.

Abbreviations: CI, confidence interval; IFN, interferon; MIRO, Markers of type I Interferon Response in Olink; NPX, normalized protein expression; UKPSSR, United Kingdom Primary Sjogren's Syndrome Registry.

We have also added a sentence to the results : “We confirmed that components of the MIRO score are ISGs and found significant correlation between the oligoprotein interferon signature and interferon- α concentrations, but not interferon- γ (Supplementary Figure 3).”

Methods for UKPSSR were also modified to include RNA data: “The UKPSSR is a national observational cohort of SjD patients fulfilling the 2002 American European Consensus Group classification criteria³ and recruited across 30 UK centres with serum samples and whole blood (PAXgene tubes [Becton, Dickinson and Company]) stored at recruitment (Eppendorf tubes at -80°C)⁴. RNA was assayed with the Illumina Human HT-12 v4 Expression BeadChip (Illumina). Globin signal suppression was performed using the Affymetrix Globin reduction protocol (Affymetrix).”

Comment 5: Additionally, adding in MRI features often found in SLE does not address the limitations. There are no official MRI biomarkers for CNS SLE, so the lack of findings is not conclusive.

We acknowledge that neutral associations for MRI features is not definite proof of neutrality for the association between *TREX1* variants and lupus. These, however, were examined because of the proposed association between *TREX1* genotypes and neurological manifestations of lupus (Namjou et al, 2011).

We have provided a clarification of this in the limitations: “Finally, neurological manifestations of SLE are complex and heterogeneous and not limited to the six clinical phenotypes examined here. These manifestations

will not be fully captured by the neurological diagnoses we have used, which is an important limitation from the perspective of studying neurolupus. Brain imaging changes are also heterogeneous in SLE and in our neuroradiological analyses we have analysed both brain volumes as well as white matter hyperintensity volume. The latter is the major structural imaging abnormality in people with SLE and likely reflects accelerated microvascular disease²³, although it is not seen in all patients and is not a specific imaging marker for SLE.”

Comment 6: Finally the point that some of this cohort may be diagnosed with SLE after the bio sample was entered was not mentioned in the study.

We thank the reviewer for this clarification.

The ambispective nature of the cohort (assessing cases both retrospectively and prospectively) is described in the introduction: “Moreover, UKB participants were aged 40-69 years at enrolment in 2006-2010 and the study captures both incident and prevalent diagnoses, meaning that the large majority of participants predisposed to SLE will by now have been diagnosed, as SLE incidence peaks in middle age⁵.”

We acknowledge that the timing of blood sampling can be more explicitly stated and we have clarified this in the revised version.

We have clarified the timing of blood sampling in the first results section on proteomics: “To overcome these challenges, we therefore sought to develop and validate an oligoprotein interferon signature of type I interferon activation derived from broad-capture proteomic datasets measured at recruitment in UKB.”

We have also modified the Methods to clarify the timing of blood sampling for proteomics: “We used data on randomly selected (~47k) and consortium-chosen (~6k) participants at baseline, excluding those who took part in the COVID-19 repeat-imaging study (~1k).”

Reviewer 3

Comment 1: The authors have responded appropriately to the comments raised in my review and the revisions are acceptable from the point of view specifically of my questions.

We thank the reviewer for this positive comment.

References

1. Namjou B, Kothari PH, Kelly JA, et al. Evaluation of the TREX1 gene in a large multi-ancestral lupus cohort. *Genes Immun* 2011;12:270-279.
2. Hanly JG, Thompson K, Skedgel C. Identification of patients with systemic lupus erythematosus in administrative healthcare databases. *Lupus* 2014;23:1377-1382.
3. Vitali C. Classification criteria for Sjogren's syndrome: a revised version of the European criteria proposed by the American-European Consensus Group. *Annals of the Rheumatic Diseases* 2002;61:554-558.
4. Ng W-F, Bowman SJ, Griffiths B, on behalf of the Usg. United Kingdom Primary Sjögren's Syndrome Registry—a united effort to tackle an orphan rheumatic disease. *Rheumatology* 2011;50:32-39.
5. Rees F, Doherty M, Grainge M, Davenport G, Lanyon P, Zhang W. The incidence and prevalence of systemic lupus erythematosus in the UK, 1999-2012. *Ann Rheum Dis* 2016;75:136-141.

Principal funder:

Medical
Research
Council

**Oligoprotein type I interferon signatures, but not *TREX1* variants,
increase risk of systemic lupus erythematosus in UK Biobank**

Dear Ching-yu and editorial team,

Many thanks for the positive response to our revised manuscript. We are pleased that the reviewers felt our manuscript was improved. Please find below a point-by-point rebuttal to the constructive comments of reviewer #2. We hope this addresses the concerns.

Colour code

Black = reviewer comments

Purple = author response

Green = new text in revised manuscript

Reviewer 1

Comment 1: The revisions address my concerns and the manuscript is now acceptable.

We thank the reviewer for the positive feedback.

Reviewer 2

Comment 1: The authors have revised the manuscript, which I still believe has merit and has findings that contribute to the body of literature on the role of *TREX1* variants in SLE. However, I find that there continue to be issues with clear limitations to the manuscript which have not been fully addressed. The language used, both in the title and throughout the paper, overstates the results with the limited data and cross-sectional design of the study. The SLE diagnosis from large data sets is not simply subject to misclassification bias. The cross-sectional design is the largest limitation here. We have no way of knowing that the subject without an ICD code for SLE at the time of the data pull did not go on to develop SLE 6 months or a year after. To state that most patients are in their 40-60s and most lupus patients present before that not sufficient. Patients present at a range of ages and we cannot make assumptions based on missing data. I would remove that statement that most patients will have been diagnosed, and simply state that you have cross sectional samples and the lack of prospective data for risk prediction is a limitation.

Cross-sectional design and case ascertainment:

We acknowledge that some participants may be diagnosed with SLE after the latest data linkage of UK Biobank to administrative healthcare datasets. We cannot exclude that a small portion of participants classified as 'non-SLE' might have gone to develop SLE, but we believe this is **unlikely to alter our conclusions** for three reasons.

First, UK Biobank is an ambispective cohort study with both cross-sectional questionnaires at recruitment (for self-reported prevalent conditions) and data linkage to administrative healthcare datasets to capture both prevalent (prior to recruitment) and incident (after recruitment) conditions. As such, UK Biobank is not limited to cross-sectional assessments at the time of recruitment. Incident SLE diagnoses in the ~15 years following the recruitment visit are also captured in the cohort.

Second, the median age at enrolment in UK Biobank was 58 years in 2006-2010, meaning that UK Biobank participants now have a median age of >70 years. Because SLE incidence peaks in middle age, we believe that most participants with a genetic predisposition to lupus should by now have been diagnosed in the dataset we used (released in AUG/2023).

Principal funder:

Medical
Research
Council

Third, we used proteomic type I interferon signatures (MIRO score) to explore the association of *TREX1* variants with immune endotypes of SLE. Although we observed strong positive associations between higher MIRO scores and the risk of incident SLE, we did not observe an association between MIRO scores and *TREX1*. This suggests that *TREX1* variants are not associated with the type I interferon-related proteomic signature observed in people predisposed to SLE.

However, we acknowledge this limitation and have made the following modification to the Introduction: “Moreover, UKB participants were aged 40-69 years at enrolment in 2006-2010 and the study captures both incident and prevalent diagnoses, meaning that ~~the large majority of most~~ participants predisposed to SLE will by now have been diagnosed, as SLE incidence peaks in middle age¹.”

Overinterpretation of results:

We agree with the reviewer that the results should be interpreted with caution to avoid overstating the findings and to ensure that the conclusions are scientifically rigorous and well supported.

Regarding the title, we believe it accurately and descriptively reflects the main study findings. Specifically, it conveys that we developed oligoprotein type I interferon signatures associated with incident SLE, while *TREX1* variants were not associated with this risk in the UK Biobank cohort. We do not feel that this interpretation is controversial or overly bold given the results obtained. We acknowledge, however, that it is difficult to capture all nuances in a title and would be happy to alter upon editorial request.

We have tempered some of the statements in the manuscript:

Introduction: “This association between *TREX1* and manifestations of SLE was considered to be confirmed following a large candidate gene study², contributing to the ~~widespread general~~ acceptance that *TREX1* variants predispose to non-monogenic SLE, which includes annotation of variants in clinical reference databases³⁻⁷.”

Introduction: “We first aimed to replicate results from a ~~canonical previous~~ genetic association study⁸ linking monoallelic *TREX1* variants to SLE.”

Results: “In summary, this systematic review and meta-analysis does not support a ~~clinically relevant strong~~ role of *TREX1* variants in SLE ~~and suggests earlier studies might have been confounded by population stratification~~”

Results: “Based on these findings, we next explored whether confounding by population stratification might partly explain ~~prior positive associations previously reported strong positive associations~~”⁸.”

Results: “Three of the 10 ~~previously reported risk variants found in cases from that study~~ were >40x more frequent in the African/African American as compared to the Non-Finnish European subpopulation of the Genome Aggregation Database (gnomAD) exome.”

Results: “Finally, no variants were observed in the family-based Finnish cohort ~~included in this first publication which used a within-family design that inherently controlled for population stratification~~ controlling for population stratification and included in a previous *TREX1*-SLE study⁸.”

Discussion: “These neutral results are consistent with prior genetic association studies retrieved through a systematic review of the literature, which together with UKB data do not support the reputed ~~strong~~ risk of SLE conferred by rare *TREX1* variants.”

Comment 2: Page 9 line 174: oligoprotein is misspelled.

Thank you. It has been corrected.

Comment 3: I agree with reviewer 1 on the dilution of variants in burden testing.

Thank you for this comment, which we addressed in the previous revision to reviewer 1's satisfaction.

Comment 4: I thank the reviewers for the detailed review of the MIRO testing. I think that validating a set of protein biomarkers for the first time in a manuscript requires a lot of effort with a high burden of proof, as they want to be using a validated test to make a conclusion. The authors have done a lot to provide evidence that MIRO is measuring Type I IFN.

We thank the reviewer for this positive comment.

Comment 5: I think the neurologic association study also remains a weak portion of the paper. I recommend the authors remove this section, as there are no validated MRI biomarkers in SLE, and more advanced MRI technologies are correlated with SLE CNS findings. An ICD code for 6 common associations in SLE can miss many other features of neuroSLE. The authors should state in the limitations that there are no official MRI biomarkers for CNS SLE, so the lack of findings is not conclusive. Conventional MRI is not very sensitive for SLE brain changes. Conventional structural MRI abnormalities, such as white matter (WM) hyperintensities, gross brain tissue atrophy and ventricular enlargement in response to this atrophy are not always observed in patients with NPSLE. In contrast, more advanced postprocessing techniques enable the quantification of structural brain metrics beyond total tissue volumes from standard T1-weighted MRI, such as regional volumes, surface area and cortical grey matter (GM) thickness. Furthermore, other modalities such as diffusion MRI, have become more useful current NPSLE clinical research. Altered tissue microstructure has been reported in several brain regions of patients with NPSLE diagnosis when compared with healthy controls and they have correlated with higher cognitive dysfunction. If these limitations can be clearly stated in the manuscript, I believe it is suitable for publication.

Thank you for these comments.

Brain imaging:

We acknowledge the limitations of brain imaging data available in UK Biobank, and that the MRI-derived phenotypes used in the cohort (white matter hyperintensity volume⁹, total brain volume, and regional brain volumes) are not the most sensitive biomarkers for diagnosing neuropsychiatric SLE (NPSLE).

Nevertheless, individuals with SLE are known to experience accelerated microvascular brain disease - typically manifested as white matter hyperintensities - and brain atrophy¹⁰, both of which are captured by these MRI measures. Our objective was not to classify participants as having or not having NPSLE within UK Biobank, but rather to assess whether *TREX1* variants are associated with MRI-derived markers relevant to SLE-related brain involvement.

In our limitations we have added substantial caveats that our approach to quantifying brain disease in lupus has significant limitations. We have added the following comments in light of the reviewer suggestions:

However as yet there is no reliable neuroimaging biomarker for neurolupus, and therefore the absence of an association is not conclusive.

We have also modified the limitations section to better reflect the value of brain imaging in the study: "Brain imaging changes are also heterogeneous in SLE and in our neuroradiological analyses we have analysed both brain volumes as well as white matter hyperintensity volume. The latter is a the major structural imaging abnormality seen in people with SLE and likely that reflects accelerated microvascular disease⁹, although it is not seen in all patients and is not a specific imaging marker for SLE."

Principal funder:

Medical
Research
Council

References

1. Rees F, Doherty M, Grainge M, Davenport G, Lanyon P, Zhang W. The incidence and prevalence of systemic lupus erythematosus in the UK, 1999-2012. *Ann Rheum Dis* 2016;75:136-141.
2. Namjou B, Kothari PH, Kelly JA, et al. Evaluation of the TREX1 gene in a large multi-ancestral lupus cohort. *Genes Immun* 2011;12:270-279.
3. Rice GI, Rodero MP, Crow YJ. Human disease phenotypes associated with mutations in TREX1. *J Clin Immunol* 2015;35:235-243.
4. Mohan C, Putterman C. Genetics and pathogenesis of systemic lupus erythematosus and lupus nephritis. *Nature Reviews Nephrology* 2015;11:329-341.
5. Teruel M, Alarcon-Riquelme ME. The genetic basis of systemic lupus erythematosus: what are the risk factors and what have we learned. *J Autoimmun* 2016;74.
6. Kaul A, Gordon C, Crow MK, et al. Systemic lupus erythematosus. *Nature Reviews Disease Primers* 2016;2:16039.
7. Caielli S, Wan Z, Pascual V. Systemic lupus erythematosus pathogenesis: interferon and beyond. *Annu Rev Immunol* 2023;41.
8. Lee-Kirsch MA, Gong M, Chowdhury D, et al. Mutations in the gene encoding the 3'-5' DNA exonuclease TREX1 are associated with systemic lupus erythematosus. *Nature Genetics* 2007;39:1065-1067.
9. Wiseman SJ, Bastin ME, Jardine CL, et al. Cerebral Small Vessel Disease Burden Is Increased in Systemic Lupus Erythematosus. *Stroke* 2016;47:2722-2728.
10. Liu S, Cheng Y, Zhao Y, et al. Clinical Factors Associated with Brain Volume Reduction in Systemic Lupus Erythematosus Patients without Major Neuropsychiatric Manifestations. *Frontiers in Psychiatry* 2018;Volume 9 - 2018.

Principal funder:

Medical
Research
Council